# Species-specific emergence of H7 highly pathogenic avian influenza virus is driven by intrahost selection differences between chickens and ducks

**Anja C. M. de Bruin, Monique I. Spronken, Adinda Kok, Miruna E. Rosu, Dennis de Meulder, Stefan van Nieuwkoop, Pascal Lexmond, Mathis Funk, Lonneke M. Leijten, Theo M. Bestebroer, Sander Herfst, Debby van Riel, Ron A. M. Fouchier, Mathilde Richard** *

Department of Viroscience, Erasmus Medical Center, Rotterdam, The Netherlands

* m.richard@erasmusmc.nl

**Data Availability Statement:** All relevant data are within the manuscript and its Supporting Information files.

## Abstract

Highly pathogenic avian influenza viruses (HPAIVs) cause severe hemorrhagic disease in terrestrial poultry and are a threat to the poultry industry, wild life, and human health. HPAIVs arise from low pathogenic avian influenza viruses (LPAIVs), which circulate in wild aquatic birds. HPAIV emergence is thought to occur in poultry and not wild aquatic birds, but the reason for this species-restriction is not known. We hypothesized that, due to species-specific tropism and replication, intrahost HPAIV selection is favored in poultry and disfavored in wild aquatic birds. We tested this hypothesis by co-inoculating chickens, representative of poultry, and ducks, representative of wild aquatic birds, with a mixture of H7N7 HPAIV and LPAIV, mimicking HPAIV emergence in an experimental setting. Virus selection was monitored in swabs and tissues by RT-qPCR and immunostaining of differential N-terminal epitope tags that were added to the hemagglutinin protein. HPAIV was selected in four of six co-inoculated chickens, whereas LPAIV remained the major population in co-inoculated ducks on the long-term, despite detection of infectious HPAIV in tissues at early time points. Collectively, our data support the hypothesis that HPAIVs are more likely to be selected at the intrahost level in poultry than in wild aquatic birds and point towards species-specific differences in HPAIV and LPAIV tropism and replication levels as possible explanations.

## Author summary

Highly pathogenic avian influenza viruses (HPAIVs) cause severe disease in poultry with mortality rates reaching 100% and, therefore, pose a large burden on the poultry industry. Additionally, some HPAIVs have spilled back from poultry into wild bird populations, increasing their geographic spread. HPAIVs arise from low pathogenic avian influenza viruses (LPAIVs), which circulate in wild aquatic birds and occasionally spillover into poultry. LPAIV to HPAIV conversion is associated with terrestrial poultry species, but the

**Funding:** This work has received funding from the European Union's Horizon 2020 research and innovation program under DELTA-FLU, grant agreement No. 727922 (M.R, R.F, A.C.dB, https://cordis.europa.eu/project/id/727922) and NIAID/NIH, contract number HHSN272201400008C (M.R, R.F, S.H, https://govtribe.com/award/federal-contract-award/definitive-contract-hhsn272201400008c). The funders had no role in study design, data collection and analysis, decision to publish, or preparation of the manuscript.

**Competing interests:** The authors have declared that no competing interests exist.

reasons underlying this species-restriction are unknown. The second step of HPAIV emergence, following HPAIV genesis, constitutes of the intrahost selection of the HPAIV from the large pool of replicating LPAIVs. Here, we investigated whether the intrahost selection efficiency differs between chickens and ducks, models for poultry and wild aquatic birds respectively, by co-inoculating them with HPAIV and LPAIV. Tagged viruses were utilized to monitor LPAIV and HPAIV frequencies at both the RNA and protein level. The HPAIV was selected in a majority of the chickens, demonstrated by the development of canonical HPAI disease and infectious HPAIV shedding, whereas all ducks solely shed infectious LPAIV. These results confirm that intrahost selection of HPAIVs is species-specific, which likely contributes to the restriction of HPAIV-emergence to poultry populations.

## Introduction

Low pathogenic avian influenza viruses (LPAIVs) are enzootic in wild aquatic birds of the orders of *Anseriformes* (e.g., ducks and geese) and *Charadriiformes* (e.g., gulls) [1,2], from which they can spillover into poultry. In poultry, LPAIVs of the H5 and H7 hemagglutinin (HA) subtypes occasionally give rise to highly pathogenic avian influenza viruses (HPAIVs). LPAIVs cause mild disease in chickens whereas HPAIVs cause severe systemic and hemorrhagic disease with mortality rates reaching 100% within few days [3]. This difference in virulence is determined by the proteolytic cleavage site in the HA protein [4,5]. HA is produced as precursor protein HA0 and is cleaved into the HA1 and HA2 subunits to activate the fusogenic potential required for membrane fusion during virus entry into host cells [6]. LPAIVs contain a monobasic HA cleavage site, consisting of a single R or K residue, that is cleaved by tissue-restricted trypsin-like proteases and this limits virus replication to specific tissues in the respiratory and intestinal tracts of birds [3,7]. In contrast, HPAIVs contain a multibasic cleavage site (MBCS) motif, which arises through single nucleotide substitutions, duplication of neighboring sequences, and/or non-homologous recombination with host or viral RNA [8,9]. The canonical MBCS consensus motif is R-X-R/K-R, which is cleaved by ubiquitously expressed furin-like proteases that allow for HA activation and thus virus replication in a plethora of different tissues in chickens [10,11].

The acquisition of an MBCS and the ensuing emergence of an HPAIV has been documented 51 times in nature, from which at least 45 were linked to terrestrial poultry holdings (primarily chickens, turkeys, and ostriches) (reviewed in [8]). Direct evidence that the conversion of LPAIV to HPAIV occurred in poultry species is present in 12 cases, in which the low pathogenic progenitor was detected in poultry prior to HPAIV emergence [8,12–18]. Although LPAIVs circulate extensively in wild aquatic birds, newly converted HPAIVs have rarely been detected in these species [8]. The reason underlying the apparent restriction of HPAIV emergence to terrestrial poultry populations is still unknown.

In terrestrial poultry, HPAIVs cause severe systemic disease due to HA cleavage by ubiquitous proteases and their ability to replicate in endothelial cells [19–22]. In contrast, most species of domestic and wild duck infected with classical H5 or H7 HPAIVs develop no or mild disease and viral antigen expression is restricted to the epithelial cells of the respiratory and intestinal tracts [23,24], although viral RNA and occasional infectious virus can be detected outside those tissues [24–27]. Only H5 HPAIVs from the A/goose/Guangdong/1/96 (Gs/Gd) lineage that have adapted to replicate in many different bird species can cause severe and systemic disease characterized by a broader tissue and cell type tropism [28–30], and are therefore

not representative of newly converted HPAIVs. Still, virtually all HPAIVs, including those from the Gs/Gd-lineage, are not endotheliotropic in most species of wild aquatic bird [31].

The emergence of novel HPAIVs can be divided into three steps. Firstly, an LPAIV acquires an MBCS. Secondly, this newly converted HPAIV has to compete with and be selected from a much larger pool of replicating LPAIVs within the infected host (intrahost selection). Thirdly, the HPAIV has to be transmitted efficiently amongst a population of naïve hosts (interhost selection). Little is known about the factors driving each of these three steps and how they might govern the restriction of HPAIV emergence to terrestrial poultry. Here, we hypothesize that the intrahost selection of HPAIVs is favoured in poultry but not in wild birds and that this is determined by intrinsic differences in HPAIV tropism, ability to efficiently spread systemically, and viral replication levels between bird species. To test this hypothesis, we qualitatively studied HPAIV selection from a larger pool of low pathogenic precursors in chickens (*Gallus gallus domesticus*) and ducks (*Anas platyrhynchos domesticus*), which were used as model species for terrestrial poultry and wild aquatic birds respectively. Birds were mono- or co-inoculated with ~$10^4$ EID$_{50}$ HPAIV, a dose that would likely result in productive infection [24,32], and a 100-fold higher dose of ~$10^6$ EID$_{50}$ LPAIV and selection was monitored over time. To represent the classical process of HPAIV emergence, characterized by LPAIV to HPAIV conversion following LPAIV spillover from wild birds to poultry, we performed the experiments with a virus pair based on the H7N7 HPAIV A/chicken/Netherlands/1/03. This HPAIV was isolated from the index farm during the Dutch outbreak in 2003 and no prior LPAIV circulation in poultry was reported nor were adaptations to poultry, such as neuraminidase stalk deletions, identified in the viral genome [33]. In addition to virus discrimination based on RNA, we engineered viruses with N-terminal epitope tags that allowed a virus-specific determination of infectious virus levels and location of virus replication in tissues. Our results demonstrate that HPAIVs have a greater selective advantage over LPAIVs in chickens than in ducks and that species-specific factors such as virus replication efficiency and tropism influence the selection of HPAIVs in an individual host.

## Results

### Epitope tagged LPAIVs and HPAIVs are not attenuated in vitro and in vivo

To study the intrahost competition between HPAIV and LPAIV in chickens and ducks, a pair of viruses based on the H7N7 HPAIV A/chicken/Netherlands/1/03 was generated by reverse genetics. As a direct progenitor of the H7N7 HPAIV (H7N7-HP) has not been identified, the LPAIV (H7N7-LP) in this study was generated artificially by mutating the MBCS PKRRRR*G to the Eurasian H7 LPAIV cleavage site nucleotide consensus sequence, resulting in monobasic motif PKGR*G. For the distinction between HPAIV and LPAIV at the protein level, N-terminal HA and FLAG epitope tags were added to the HA protein (Fig 1A). H7N7-LP was equipped with an eight amino acid FLAGtag (H7N7-LP$_{FLAGtag}$) and H7N7-HP with a nine amino acid HAtag (H7N7-HP$_{HAtag}$) (Fig 1A). As control, H7N7-LP was equipped with HAtag as well (H7N7-LP$_{HAtag}$). The tags were placed after the signal peptide and connected to the mature HA via a GGGGS linker sequence as described previously [34]. The expression and detection of HAs with epitope tag were confirmed by flow cytometry upon inoculation of a chicken fibroblast cell line (DF-1 cells) with H7N7-LP$_{FLAGtag}$ or H7N7-HP$_{HAtag}$, although the epitope tag staining was less sensitive than influenza virus nucleoprotein (NP) staining (Fig 1B). No attenuation was observed when comparing multi-cycle replication kinetics between wild-type and epitope tagged virus upon inoculation of primary chicken or duck embryonic fibroblasts (CEF; DEF) at a multiplicity of infection (MOI) of 0.001 (Fig 1C and 1D).

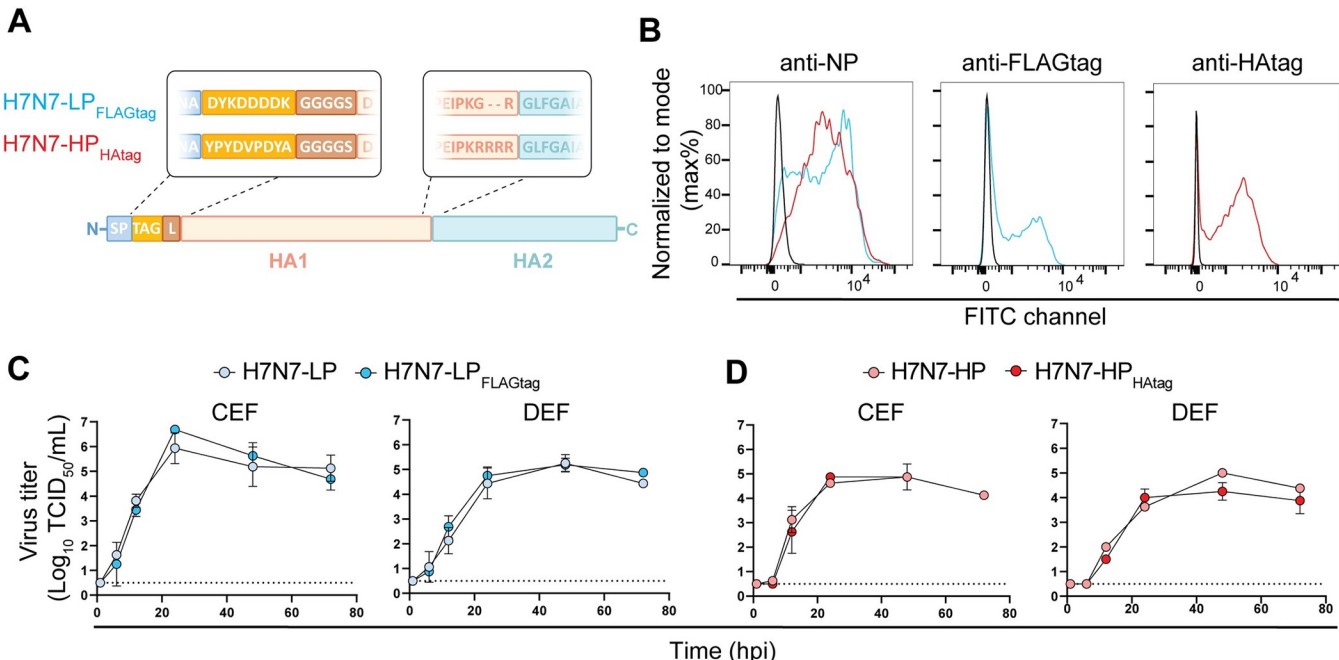

**Fig 1. Similar in vitro replication kinetics of wild-type and epitope tagged H7N7 LPAIVs and HPAIVs.** (A) Schematic representation of the modified HA proteins of H7N7 LPAIVs and HPAIVs that were used in this study. The FLAGtag and HAtag are located downstream of the signal peptide (SP) and are connected to the mature HA1 via a linker (L) sequence. The viruses differ only at the N-terminus and at the cleavage site at the border of HA1 and HA2. (B) Chicken fibroblasts (DF-1 cells) were inoculated with H7N7-LP$_{FLAGtag}$ (in blue), H7N7-HP$_{HAtag}$ (in red), or mock-inoculated (in black) at an MOI of 0.1. Expression of the influenza virus nucleoprotein (NP), FLAGtag, and HAtag was assessed at 24 hpi by flow cytometry. (C, D) Replication kinetics in primary chicken and duck embryonic fibroblasts (CEF; DEF) following inoculation with H7N7-LP, H7N7-LP$_{FLAGtag}$, H7N7-HP, or H7N7-HP$_{HAtag}$ at an MOI of 0.001. Infectious titers in supernatants were determined by endpoint titration in MDCK cells and expressed as log$_{10}$ TCID$_{50}$/mL. Data are presented as arithmetic mean ± SD of log$_{10}$ transformed values from two independent experiments. The dotted lines indicate the limit of detection of the endpoint titration assay.

Next, in vivo infection dynamics of H7N7-LP in domestic ducks were compared to those of H7N3-LP A/mallard/Netherlands/12/00, to assess whether H7N7-LP is representative of natural mallard LPAIVs. Six ducks per group were inoculated intratracheally and intra-oesophageally with ~10$^6$ EID$_{50}$ of virus and viral shedding levels, body weight, and disease severity were determined daily (S1A and S1B Fig). At 3 days post inoculation (dpi) and 7 dpi, three animals were euthanized for virological examination of the tissues. None of the LPAIV-inoculated ducks showed clinical signs and the ducks increased in body weight over time (S2A Fig). Similar levels of oropharyngeal (OP) and cloacal (CL) viral RNA shedding, viral RNA levels in tissues, and detection of viral RNA in the drinking water (provided twice daily in buckets) were observed in the H7N3-LP and H7N7-LP inoculated groups (S3A, S3B, S3E and S4A Figs), indicating that H7N7-LP is representative of a mallard LPAIV. Secondly, infection dynamics of H7N7-LP, H7N7-LP$_{FLAGtag}$, and H7N7-LP$_{HAtag}$ were compared in ducks to determine whether the choice and presence of an epitope tag influences LPAIV replication in vivo. Again, similar levels of OP and CL viral RNA shedding, and viral RNA detection in tissues and drinking water were observed in all three groups (S3B–S3E and S4A Figs), indicating the absence of attenuation caused by a tag.

Finally, infection dynamics of H7N7-HP and H7N7-HP$_{HAtag}$ were compared in chickens to determine whether the presence of an epitope tag influences HPAIV replication in vivo. Chickens were inoculated with the low dose of ~10$^4$ EID$_{50}$ of virus, because the competition experiments were planned to be performed with ~10$^4$ EID$_{50}$ HPAIV and ~10$^6$ EID$_{50}$ LPAIV. Comparable disease progression, shedding of viral RNA, and detection of viral RNA in tissues

were observed in the two groups, except for chicken C11 in the H7N7-HP$_{HAtag}$ group which showed a multiple day delay in disease progression and shedding (S5 Fig). Clinical manifestations included a decrease in weight gain or even weight loss, lethargy, edema of the head, wattle, and comb, and eventually hemorrhaging on the comb and feet (S2B and S5A Figs). Interestingly, only high levels of H7N7-HP$_{HAtag}$ RNA were detected in the drinking water (S4B Fig). Collectively, the presence of an HAtag did not substantially attenuate H7N7-HP$_{HA-tag}$ and did not prevent the development of HPAI disease in chickens.

## Differential infection phenotypes of LPAIV and HPAIV in chickens and ducks

As the presence of epitope tags in the HA protein did not markedly attenuate the H7N7 viruses, we continued with H7N7-HP$_{HAtag}$ and H7N7-LP$_{FLAGtag}$ and characterized their infection dynamics at the infectious virus level in both chickens and ducks.

Chickens inoculated with ~$10^4$ EID$_{50}$ of H7N7-HP$_{HAtag}$ shed high amounts of viral RNA and infectious virus via both the OP and CL routes (Figs 2A, 2E and S6A and S6B (for shedding of individual animals)). The ubiquitous detection of H7N7-HP$_{HAtag}$ RNA in tissues from all euthanized chickens was supported by high infectious virus levels, which confirmed the expected systemic dissemination of the H7N7-HP$_{HAtag}$ (Figs 3A, 3C and S7A, S7C). In contrast to H7N7-HP$_{HAtag}$, chickens inoculated with ~$10^6$ EID$_{50}$ of H7N7-LP$_{FLAGtag}$ did not show any clinical signs throughout the experiment and gained weight (S2B Fig). H7N7-LP$_{FLAGtag}$ RNA

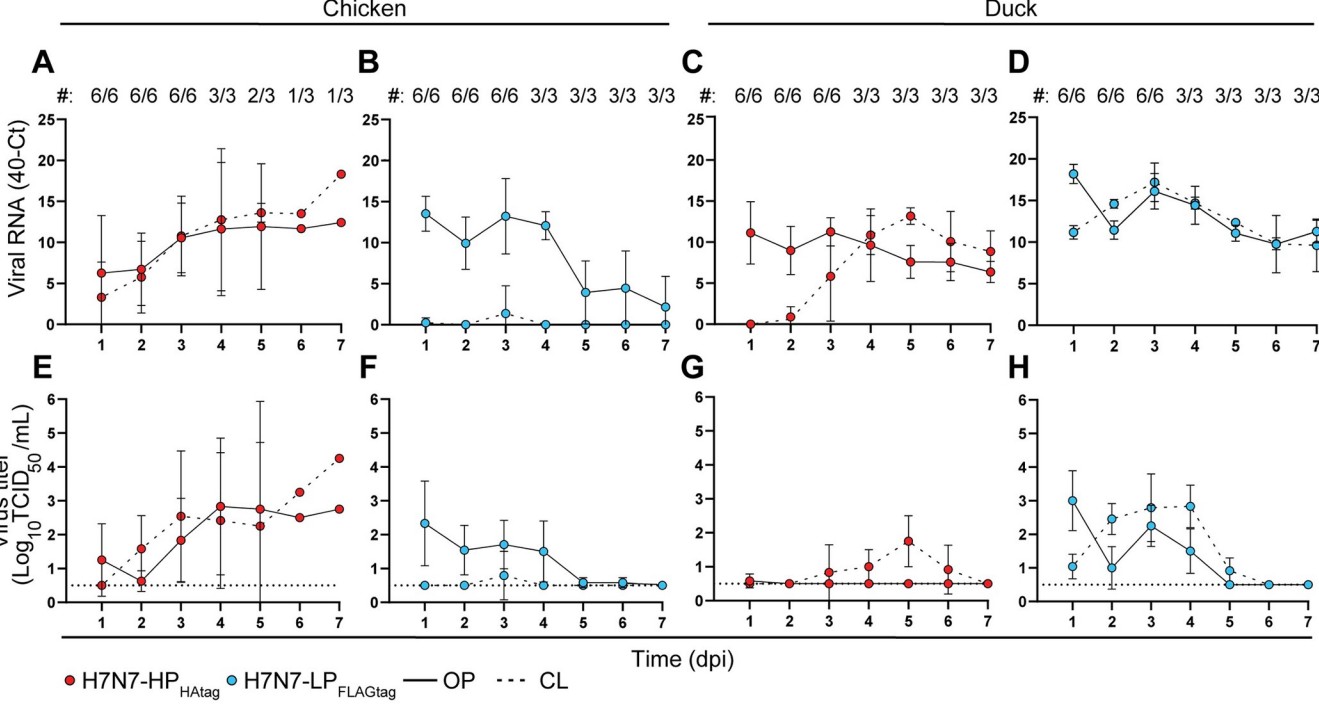

**Fig 2. Shedding of H7N7-HP$_{HAtag}$ and H7N7-LP$_{FLAGtag}$ in chickens and ducks.** (A-D) Viral RNA quantification in oropharyngeal (OP) and cloacal (CL) swabs of chickens and ducks mono-inoculated with ~$10^4$ EID$_{50}$ of H7N7-HP$_{HAtag}$ or ~$10^6$ EID$_{50}$ of H7N7-LP$_{FLAGtag}$. Viral RNA levels were determined by RT-qPCR targeting the influenza virus matrix segment and expressed as 40-cycle threshold (Ct). Data are depicted as arithmetic mean ± SD calculated from the number of animals indicated above the graphs (#: the number of alive animals/the total number of animals in the experiment). (E-H) Infectious virus titers in swabs from panels (A-D). Infectious titers were determined by endpoint titration in MDCK cells and expressed as log$_{10}$ TCID$_{50}$/mL. Data are depicted as arithmetic mean ± SD calculated from log$_{10}$ transformed values from the number of animals indicated above panels (A-D). The horizontal dotted lines indicate the limit of detection of the endpoint titration assay.

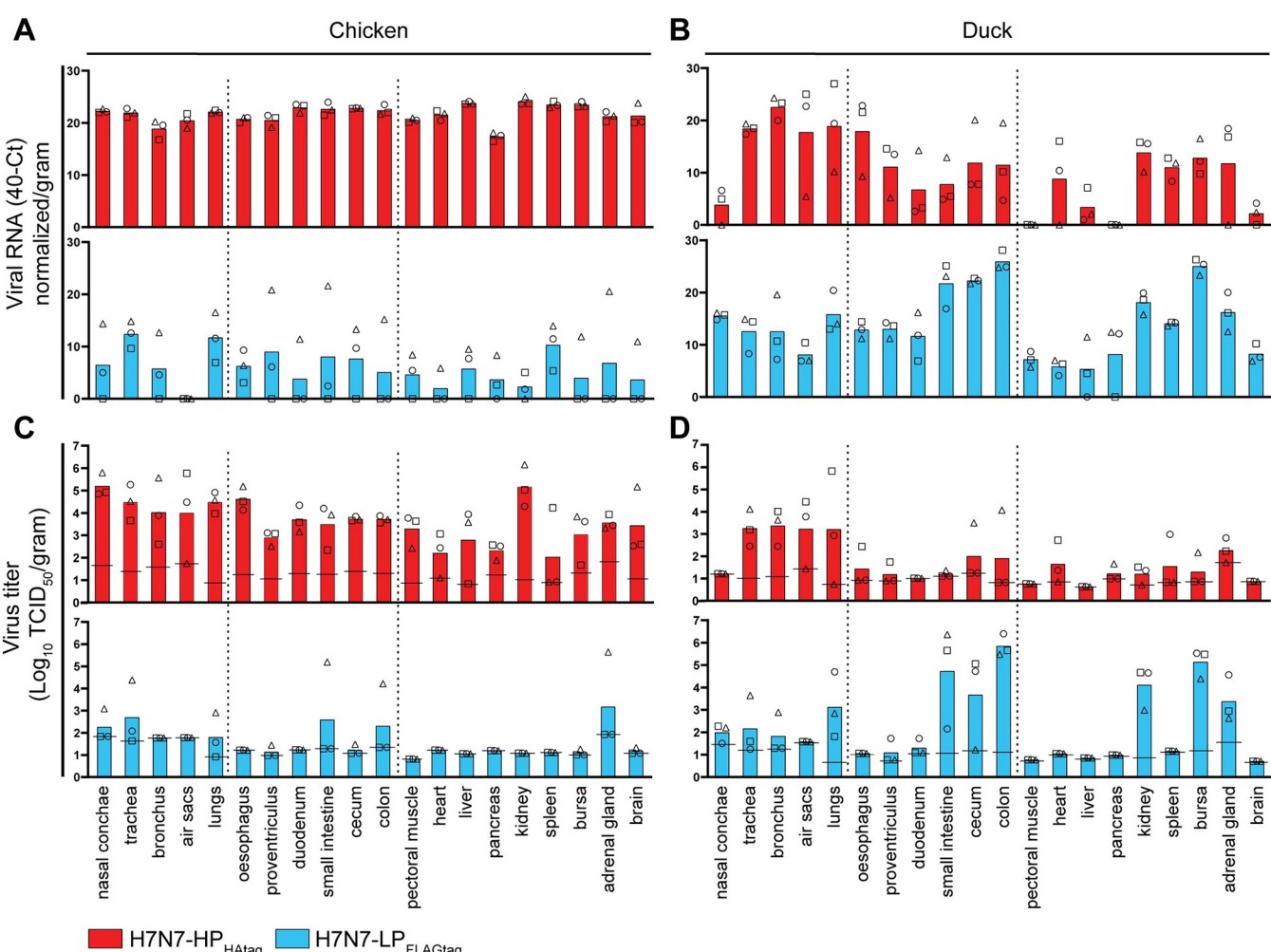

**Fig 3. Detection of H7N7-HP_{HAtag} and H7N7-LP_{FLAGtag} in tissues of chickens and ducks at 3 dpi.** (A, B) Viral RNA quantification in tissues harvested at 3 dpi from chickens (A) and ducks (B) inoculated with ~$10^4$ EID$_{50}$ of H7N7-HP$_{HAtag}$ or ~$10^6$ EID$_{50}$ of H7N7-LP$_{FLAGtag}$. Viral RNA amounts were determined by RT-qPCR targeting the influenza virus matrix segment and expressed as 40-cycle threshold (Ct) normalized/gram tissue. Bars represent the arithmetic mean. The dotted lines distinguish tissues from the respiratory, digestive, and miscellaneous systems. (C, D) Infectious virus titers in tissues from panels (A, B). Infectious titers were determined by endpoint titration in MDCK cells and expressed as log$_{10}$ TCID$_{50}$/gram tissue. Bars represent the arithmetic mean of log$_{10}$ transformed values. Dotted lines similar to (A, B). The horizontal solid lines indicate the limit of detection of the endpoint titration assay per tissue.

and infectious virus were detected in OP swabs, but rarely or not in CL and drinking water swabs respectively (Figs 2B, 2F, S4B, S4D, S6A and S6B). Although viral RNA was detected in multiple tissues at 3 and 7 dpi, infectious virus was only detected in tissues from the intestinal and respiratory tracts of one of the chickens at 3 dpi (Fig 3A and 3C) and in none of the chickens at 7 dpi (S7A and S7C Fig). In conclusion, H7N7-HP$_{HAtag}$ replicated to higher levels and in a broader range of chicken tissues and was shed more than H7N7-LP$_{FLAGtag}$, especially via the CL route.

In ducks, H7N7-LP$_{FLAGtag}$ RNA and infectious virus were shed for seven and four days respectively (Figs 2D, 2H and S6C, S6D (for shedding of individual animals)) and were detected in the drinking water, which was visually contaminated with fecal excretions (S4A and S4C Fig). At 3 dpi, H7N7-LP$_{FLAGtag}$ RNA and infectious virus were primarily detected in the lung, small intestine, cecum, colon, bursa, and kidney (Fig 3B and 3D). At 7 dpi, viral RNA but no infectious virus was detected in the cecum, colon, and bursa (S7B Fig). Upon

mono-inoculation of ducks with ~$10^4$ $EID_{50}$ of H7N7-HP$_{HAtag}$, ducks did not show clinical signs and body weights increased over time (S2A Fig). Viral RNA was detected in OP swabs on all seven days, whereas CL shedding of viral RNA started at 3 or 4 dpi (Figs 2C and S6C). Interestingly, no infectious virus was recovered from the OP and drinking water swabs, while the CL swabs contained low infectious virus levels (Figs 2G, S4A, S4C and S6D). The respiratory tract was the primary site of viral replication of H7N7-HP$_{HAtag}$ at 3 dpi with highest RNA and infectious virus levels in the bronchus, trachea, air sacs, and lungs (Figs 3B, 3D, S6C and S6D). At 7 dpi, H7N7-HP$_{HAtag}$ RNA was present in the bronchus, oesophagus, small intestine, cecum, and colon (S7B Fig), but no infectious virus was detected (S7D Fig). Collectively, in contrast to inoculated chickens, H7N7-LP$_{FLAGtag}$ replicated to higher levels in tissues from ducks and was shed more than H7N7-HP$_{HAtag}$.

## Intrahost selection of H7N7-HP$_{HAtag}$ in chickens co-inoculated with H7N7-HP$_{HAtag}$ and H7N7-LP$_{FLAGtag}$

We continued with the examination of the intrahost selection of a minority of H7N7-HP$_{HAtag}$ from a large pool of H7N7-LP$_{FLAGtag}$ in chickens. To this end, six chickens were co-inoculated with a mixture of H7N7-HP$_{HAtag}$ ($10^{4.3}$ $EID_{50}$) and H7N7-LP$_{FLAGtag}$ ($10^{6.1}$ $EID_{50}$), corresponding to a 1:62 ratio rather than the intended 1:100 ratio. To determine which virus was selected in the animals, we employed an HPAIV/LPAIV differentiating RT-qPCR targeting the cleavage site. The RT-qPCR targeting the matrix segment, used during all mono-inoculation experiments, was also performed. A high correlation ($R^2$ = 0.98, slope = 0.94, intercept = 0.058) was observed between the HPAIV/LPAIV and matrix 40-Ct values and thus the remainder of the figures only show HPAIV/LPAIV 40-Ct values. Additionally, a quantitative plaque assay that employed the epitope tags placed in the HA proteins was developed that allowed allocation of each plaque to either H7N7-HP$_{HAtag}$ or H7N7-LP$_{FLAGtag}$ and consequently the determination of H7N7-HP$_{HAtag}$- and H7N7-LP$_{FLAGtag}$-specific plaque forming units (PFU) in swab and tissue samples. Due to aspecific signal of the FLAGtag staining in infected cells, plaques were immunostained for NP and HAtag; plaques positive for only NP were designated as H7N7-LP$_{FLAGtag}$-positive and plaques positive for both NP and HAtag were designated as H7N7-HP$_{HAtag}$-positive (Fig 4A). Conventional endpoint titrations in MDCK cells were also performed to compare infectious virus titers between mono- and co-inoculation experiments (S8 and S10 Figs).

At 3 dpi, clinical signs indicative of HPAIV infection, such as drowsiness and/or edema of the head, were observed in four of six chickens (Fig 4B). Assessment of the shedding data using HPAIV/LPAIV differentiating RT-qPCR and plaque assay demonstrated high H7N7-HP$_{HAtag}$ RNA and infectious virus shedding levels at 2 and 3 dpi in OP and CL swabs in these four chickens (Figs 4C, 4D and S8A). RNA and infectious virus of both H7N7-HP$_{HAtag}$ and H7N7-LP$_{FLAGtag}$ were detected in the drinking water (S9A and S9C Fig). Tissues of chicken C19, C20, C21, and C24 showed a ubiquitous presence of H7N7-HP$_{HAtag}$ RNA (Figs 4E and S10C) and infectious virus was broadly detected as well (Figs 4F, S10A and S10E). However, not all tissues of chickens euthanized at 3 dpi were positive for H7N7-HP$_{HAtag}$ and titers were generally lower than in tissues from H7N7-HP$_{HAtag}$ mono-inoculated chickens (Fig 3C), most notably in the nasal conchae, intestinal tract, pectoral muscle, and brain (Figs 4F, S10A and S10E). Interestingly, concomitant replication of H7N7-HP$_{HAtag}$ and H7N7-LP$_{FLAGtag}$ in tissues was only detected in chicken C20 (Fig 4E and 4F).

H7N7-HP$_{HAtag}$ was not selected in two of six co-inoculated chickens. Chicken C22 tested positive for H7N7-LP$_{FLAGtag}$ and H7N7-HP$_{HAtag}$ RNA in OP swabs, but only H7N7-LP$_{FLAGtag}$ infectious virus was detected in these (Fig 4C and 4D). In chicken C23, low levels of mostly

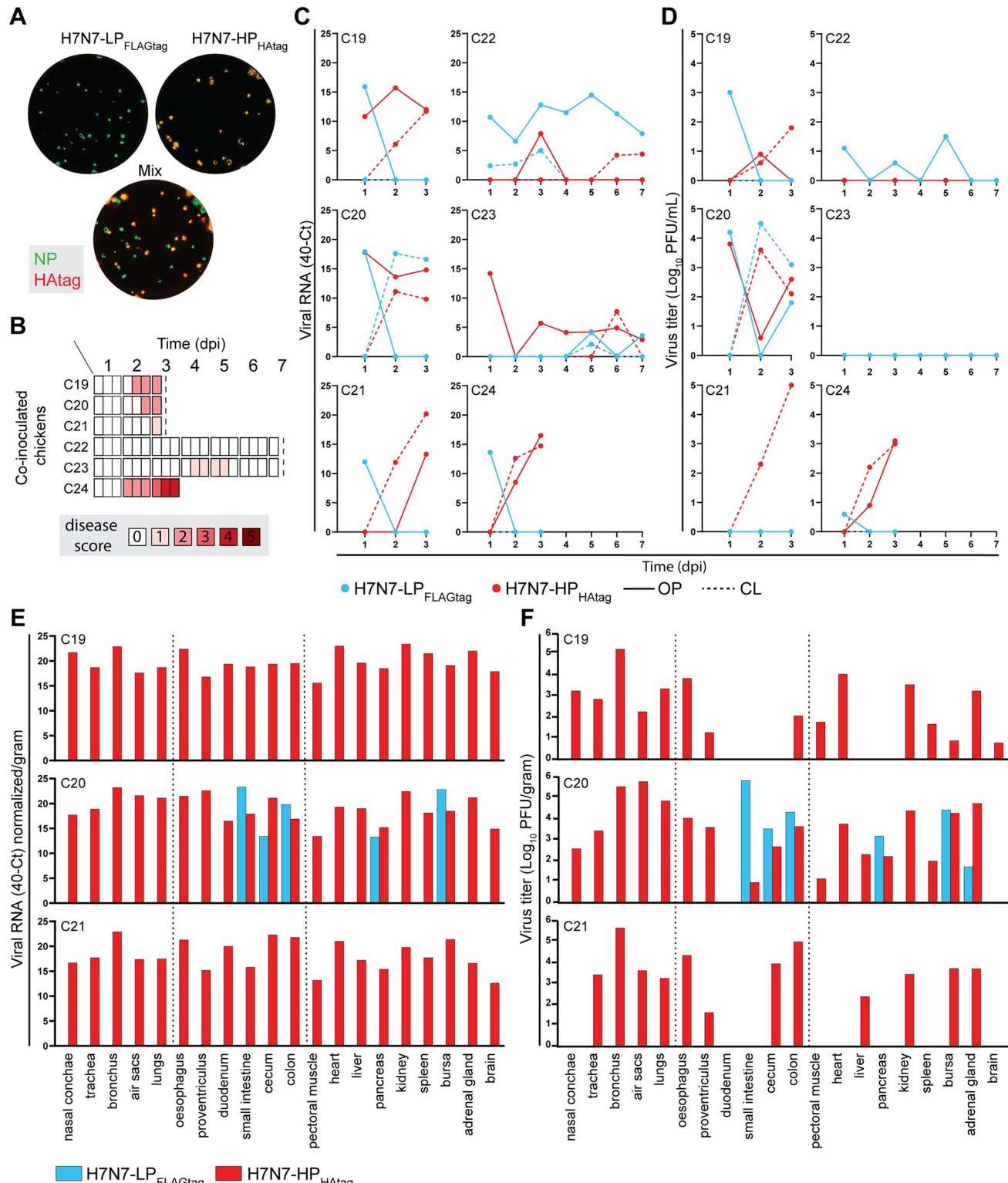

**Fig 4. Intrahost selection of H7N7-HP$_{HAtag}$ in four of six chickens co-inoculated with H7N7-HP$_{HAtag}$ and H7N7-LP$_{FLAGtag}$.** (A) Method for distinguishing H7N7-HP$_{HAtag}$ and H7N7-LP$_{FLAGtag}$ plaques based on epitope tag staining. The number of plaque forming units (PFU) per virus per sample was determined by co-staining plaques with antibodies targeting influenza virus nucleoprotein (NP) and HAtag. Plaques formed by H7N7-LP$_{FLAGtag}$ stain positive for NP (green). Plaques formed by H7N7-HP$_{HAtag}$ stain positive for both NP (green) and HAtag (red). (B) Severity of disease in six chickens co-inoculated with H7N7-HP$_{HAtag}$ and H7N7-LP$_{FLAGtag}$. Disease severity scores (0–5) were determined thrice daily, each

box representing one time point. Dotted lines indicate planned time points for euthanasia and dissection. (C) H7N7-HP$_{HAtag}$ and H7N7-LP$_{FLAGtag}$ RNA quantification in oropharyngeal (OP) and cloacal (CL) swabs from co-inoculated chickens. Viral RNA amounts were determined by HPAIV/LPAIV differentiating RT-qPCR targeting the HA cleavage site and expressed as 40-cycle threshold (Ct). (D) Infectious virus titers in the swabs from panel (C). Oropharyngeal and CL swab titers are depicted as PFU/mL as measured by the HPAIV/LPAIV differentiating plaque assay. (E) H7N7-HP$_{HAtag}$ and H7N7-LP$_{FLAGtag}$ RNA quantification in tissues harvested at 3 dpi as determined by HPAIV/LPAIV differentiating RT-qPCR targeting the HA cleavage site and expressed as 40-Ct normalized/gram tissue. The dotted lines distinguish tissues from the respiratory, digestive, and miscellaneous systems. (F) Infectious virus titers of tissues from panel (E), depicted as PFU/gram tissue as determined by the HPAIV/LPAIV differentiating plaque assay. Dotted lines are similar to (E).

H7N7-HP$_{HAtag}$ RNA but no infectious virus were detected in swabs (Figs 4C, 4D and S8A). Productive infection was likely limited in chicken C23, although some minor clinical signs, i.e., slightly hunched posture and ruffled feathers, were observed at 4 and 5 dpi (Fig 4B). Tissues from chickens C22 and C23 were harvested at 7 dpi, which was too late to detect infectious virus (S10E Fig).

## Predominant shedding of H7N7-LP$_{FLAGtag}$ from ducks co-inoculated with H7N7-HP$_{HAtag}$ and H7N7-LP$_{FLAGtag}$

Following the confirmation of the intrahost selection of the H7N7-HP$_{HAtag}$ in chickens, we continued to test our hypothesis and assessed whether HPAIV selection is disfavored in the LPAIV reservoir species. Six ducks were inoculated with the same inoculum that was used to co-inoculate the chickens, containing a 1:62 mixture of H7N7-HP$_{HAtag}$ and H7N7-LP$_{FLAGtag}$. As during the mono-inoculations, no clinical signs were observed in any of the ducks and the body weight increased during the experiment (S2C Fig). In contrast to the chickens, shedding of H7N7-LP$_{FLAGtag}$ predominated in the ducks (Fig 5A and 5B). However, H7N7-LP$_{FLAGtag}$ RNA and infectious virus shedding levels were generally lower and delayed compared to those upon mono-inoculation (Fig 2D and 2H). A small peak of H7N7-HP$_{HAtag}$ viral RNA was detected in OP swabs from five of six ducks at 2 and/or 3 dpi, but no H7N7-HP$_{HAtag}$ infectious virus was isolated from these samples in plaque assay (Fig 5A and 5B). Furthermore, the cloacal shedding observed from 4 dpi onwards in H7N7-HP$_{HAtag}$ mono-inoculated ducks (Fig 2C and 2G) was absent. Accordingly, H7N7-LP$_{FLAGtag}$ but not H7N7-HP$_{HAtag}$ RNA and infectious virus were detected in the drinking water (S9B and S9D Fig).

In contrast to the predominance of H7N7-LP$_{FLAGtag}$ in swabs from co-inoculated ducks, H7N7-HP$_{HAtag}$ RNA and infectious virus were predominant in tissues harvested at 3 dpi (Figs 5C, 5D and S10B). H7N7-HP$_{HAtag}$ RNA was present in a plethora of tissues, including some outside of the respiratory and digestive tracts, such as pectoral muscle and heart (Fig 5C). However, infectious virus was primarily recovered from respiratory and digestive tract tissues, kidney, and bursa (Figs 5D and S10B). At 7 dpi, H7N7-LP$_{FLAGtag}$ RNA was primarily detected in the intestinal tract and H7N7-HP$_{HAtag}$ RNA in the respiratory tract, but infectious virus was only recovered from duck D34 (S10D and S10F Fig). Taken together, despite detection of H7N7-HP$_{HAtag}$ in the tissues of ducks, H7N7-LP$_{FLAGtag}$ was selected on the long-term.

## Differential viral antigen staining patterns in mono- and co-inoculated chickens versus ducks

Finally, we analyzed the expression of viral antigen in tissues of mono- and co-inoculated chickens and ducks to study the differential sites of H7N7-HP$_{HAtag}$ and H7N7-LP$_{FLAGtag}$ replication, which might hold clues on the location of H7N7-HP$_{HAtag}$ intrahost selection.

In H7N7-HP$_{HAtag}$ mono-inoculated chickens euthanized at 3 dpi, NP staining was detected in a plethora of different tissues indicating systemic virus replication (Fig 6A). The comb and nasal conchae contained the highest number of NP-positive cells, which corresponds to the

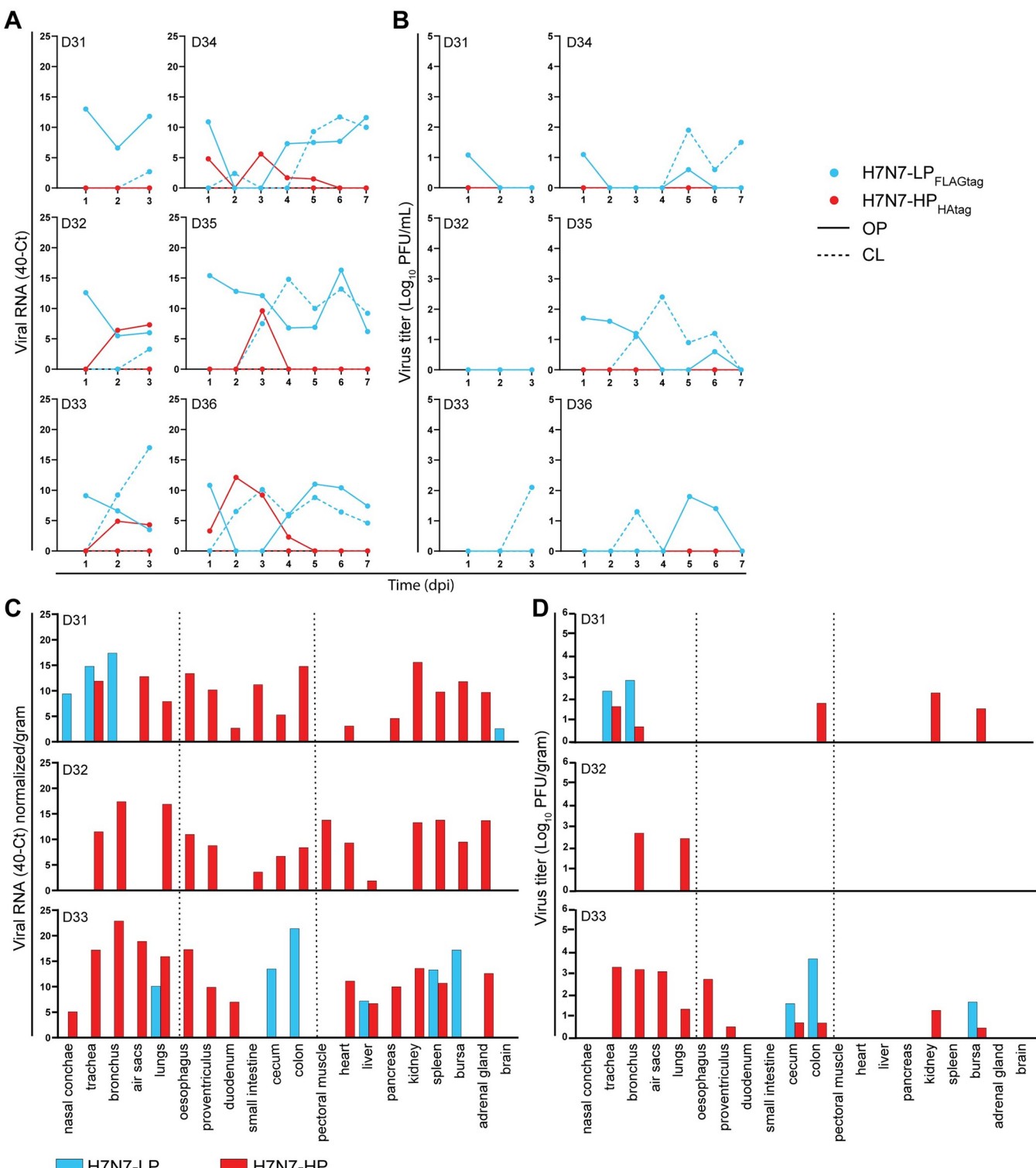

**Fig 5. H7N7-LP_FLAGtag is predominantly shed from ducks co-inoculated with H7N7-HP_HAtag and H7N7-LP_FLAGtag despite detection of H7N7-HP_HAtag in tissues at 3 dpi.** (A) H7N7-HP_HAtag and H7N7-LP_FLAGtag RNA quantification in oropharyngeal (OP) and cloacal (CL) swabs from six co-inoculated ducks. Viral RNA amounts were determined by HPAIV/LPAIV differentiating RT-qPCR targeting the HA cleavage site and expressed as 40-cycle threshold (Ct). (B) Infectious virus titers in the swabs from panel (A). Titers are depicted as PFU/mL as measured by the HPAIV/LPAIV differentiating plaque assay. (C) H7N7-HP_HAtag and H7N7-LP_FLAGtag RNA quantification in tissues harvested at 3 dpi as determined by HPAIV/LPAIV differentiating RT-qPCR targeting the HA cleavage site and expressed as 40-Ct normalized/gram tissue. The dotted lines distinguish tissues from the respiratory, digestive, and miscellaneous systems.

(D) Infectious virus titers tissues from panel (C), depicted as PFU/gram tissue as determined by the HPAIV/LPAIV differentiating plaque assay. Dotted lines are similar to (C).

tropism described for chickens naturally infected with the Dutch H7N7 HPAIVs from 2003 [21]. The cells positive for NP were identified as mostly endothelium and occasionally epithelium in the nasal conchae and kidney tubules. In contrast, the detection of NP-positive cells was limited in H7N7-LP$_{FLAGtag}$ mono-inoculated chickens in which bronchial epithelial cells (chicken C13), and tracheal and small intestinal epithelial cells and sloughed cells in the lungs (chicken C15) were NP-positive.

In ducks, NP-positivity at 3 dpi was restricted to air sac epithelial cells from H7N7-HP$_{HAtag}$ mono-inoculated duck D26 and to enterocytes in the colon of all H7N7-LP$_{FLAGtag}$ mono-inoculated ducks (Fig 6A). Other tissues were NP-negative despite the detection of viral RNA and infectious virus in many tissues, including some outside of the respiratory and intestinal tracts that suggested systemic dissemination, such as H7N7-HP$_{HAtag}$ in the heart and H7N7-LP$_{FLAGtag}$ in the kidney (Fig 3D). As a means to increase virus detection sensitivity, influenza vRNA in-situ hybridization was performed on a selection of duck tissues outside of the respiratory and intestinal tracts, but no virus infected cells were detected, despite increased detection sensitivity as observed in an NP-positive air sac (S1 Table). Thus, in contrast to the chickens, H7N7-LP$_{FLAGtag}$ NP expression was more prominent than that of H7N7-HP$_{HAtag}$ in ducks.

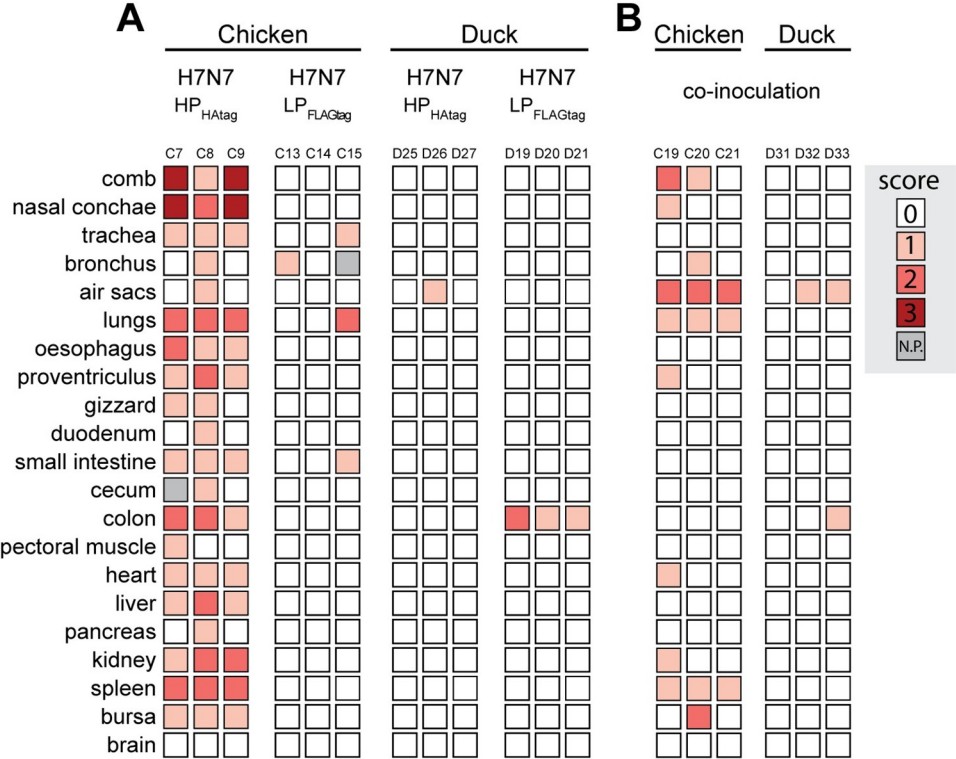

**Fig 6. Viral antigen expression in tissues of chickens and ducks mono- and co-inoculated with H7N7-HP$_{HAtag}$ and H7N7-LP$_{FLAGtag}$ at 3 dpi.** (A) Degree of viral antigen expression in tissues of chickens and ducks mono-inoculated with ~$10^4$ EID$_{50}$ H7N7-HP$_{HAtag}$ or ~$10^6$ EID$_{50}$ H7N7-LP$_{FLAGtag}$ and euthanized at 3 dpi. Viral antigen expression was determined by immunohistochemical detection of the influenza virus nucleoprotein. Tissues showing no positivity were scored as (0), sporadic single positive cells as (1), multiple sites of positive cells as (2), and diffuse positivity as (3). N.P. = not performed. (B) Degree of viral antigen expression in tissues from H7N7-LP$_{FLAGtag}$/H7N7-HP$_{HAtag}$ co-inoculated chickens and ducks euthanized at 3 dpi. Staining and scoring were performed as in (A).

Next, we assessed NP expression in H7N7-LP$_{FLAGtag}$/H7N7-HP$_{HAtag}$ co-inoculated chickens at 3 dpi (Fig 6B). The number of NP-positive tissues and NP-positive cells per tissue in co-inoculated chickens, in which H7N7-HP$_{HAtag}$ was intrahost selected, were lower than in the mono-inoculated chickens (Fig 6A). This is in accordance with infectious virus titers being detected in fewer tissues and at lower titers in the co-inoculated chickens compared to H7N7-HP$_{HAtag}$ mono-inoculated chickens (Figs 3C and 4F) and is indicative of a negative impact of the H7N7-LP$_{FLAGtag}$ co-inoculation on H7N7-HP$_{HAtag}$ replication at early time points. Chicken C24, which had reached humane endpoints at 3 dpi, showed a more widespread NP-positivity akin to one mono-inoculated chicken that was euthanized (S11 Fig). Notably, robust viral antigen expression was detected in air sac epithelial cells of the co-inoculated chickens at 3 dpi, which was mostly absent in the mono-inoculated chickens (Fig 6). No evidence for NP-positive endothelial cells in the air sacs was found. Other parts of the respiratory tract that were NP-positive included epithelial cells in the bronchus and/or nasal conchae, and single cells within air capillaries of the lung. Systemic dissemination of virus had already occurred at 3 dpi. In chicken C19, endothelial cells in the comb, proventriculus, and nasal conchae were NP-positive and virus had spread to other tissues outside of the respiratory and intestinal tracts, as evidenced by NP-positive tubular kidney epithelial cells, pericardial mesothelium, and single cells of the spleen. Extrarespiratory dissemination was detected to a lesser extent in chickens C20 and C21, in which NP-positive cells were detected in the comb and spleen or only spleen respectively.

Tissues from co-inoculated chickens that were harvested at 3 dpi and that displayed multiple sites of NP-positive cells (scored '2' in Fig 6) were analyzed with an HPAIV/LPAIV differentiating immunostaining. HAtag and NP immunostainings were performed on consecutive slides: areas positive for NP and HAtag were designated as infected by H7N7-HP$_{HAtag}$ and areas positive for NP only were designated as likely infected by H7N7-LP$_{FLAGtag}$. As controls, the NP-positive combs of H7N7-HP and H7N7-HP$_{HAtag}$ mono-inoculated chickens were included in the analysis, which stained HAtag-negative (Fig 7A) and HAtag-positive (Fig 7B) respectively. In the air sac epithelial cells of all co-inoculated chickens and the comb of chicken C19, NP expression co-localized with HAtag expression, confirming infection with H7N7-HP$_{HAtag}$ (Fig 7C–7E, 7G and 7H). In chicken C20, both HAtag-positive and HAtag-negative NP-positive epithelial cells were detected in the bursa, indicating concomitant H7N7-LP$_{FLAGtag}$ and H7N7-HP$_{HAtag}$ infection (Fig 7F and 7H). Collectively, the co-inoculated chickens presented with a comparable but delayed pattern of H7N7-LP$_{FLAGtag}$ and H7N7-HP$_{HAtag}$ antigen expression, except for the striking positivity of the air sacs for H7N7-HP$_{HAtag}$.

In co-inoculated ducks, NP expression was detected in a few cells in the air sacs and colon (Fig 6B), which constituted of too few cells to robustly perform the HAtag immunostaining. The air sacs from duck D33 contained positive epithelial cells, which were most likely infected by H7N7-HP$_{HAtag}$ as detected by the HPAIV/LPAIV differentiating RT-qPCR and plaque assay (Fig 5C and 5D). Based on those same virus detection data, the NP-positive enterocytes from the colon of duck D33 were most likely infected by H7N7-LP$_{FLAGtag}$. Again, duck tissues outside of the respiratory and intestinal tracts in which H7N7-HP$_{HAtag}$ RNA and/or infectious virus were detected were NP-negative. Therefore, influenza vRNA in-situ hybridization was performed on a subset of those tissues, but they were vRNA-negative (S1 Table). In the fixed kidney sample from duck D32, a vRNA-positive air sac was detected that was attached to the kidney, which indicated that some of the positive virology data outside of the respiratory and intestinal tracts could be due to cross-contamination by co-sampled air sacs. Collectively, H7N7-LP$_{FLAGtag}$ and H7N7-HP$_{HAtag}$ were likely replicating in the GIT and RT, respectively.

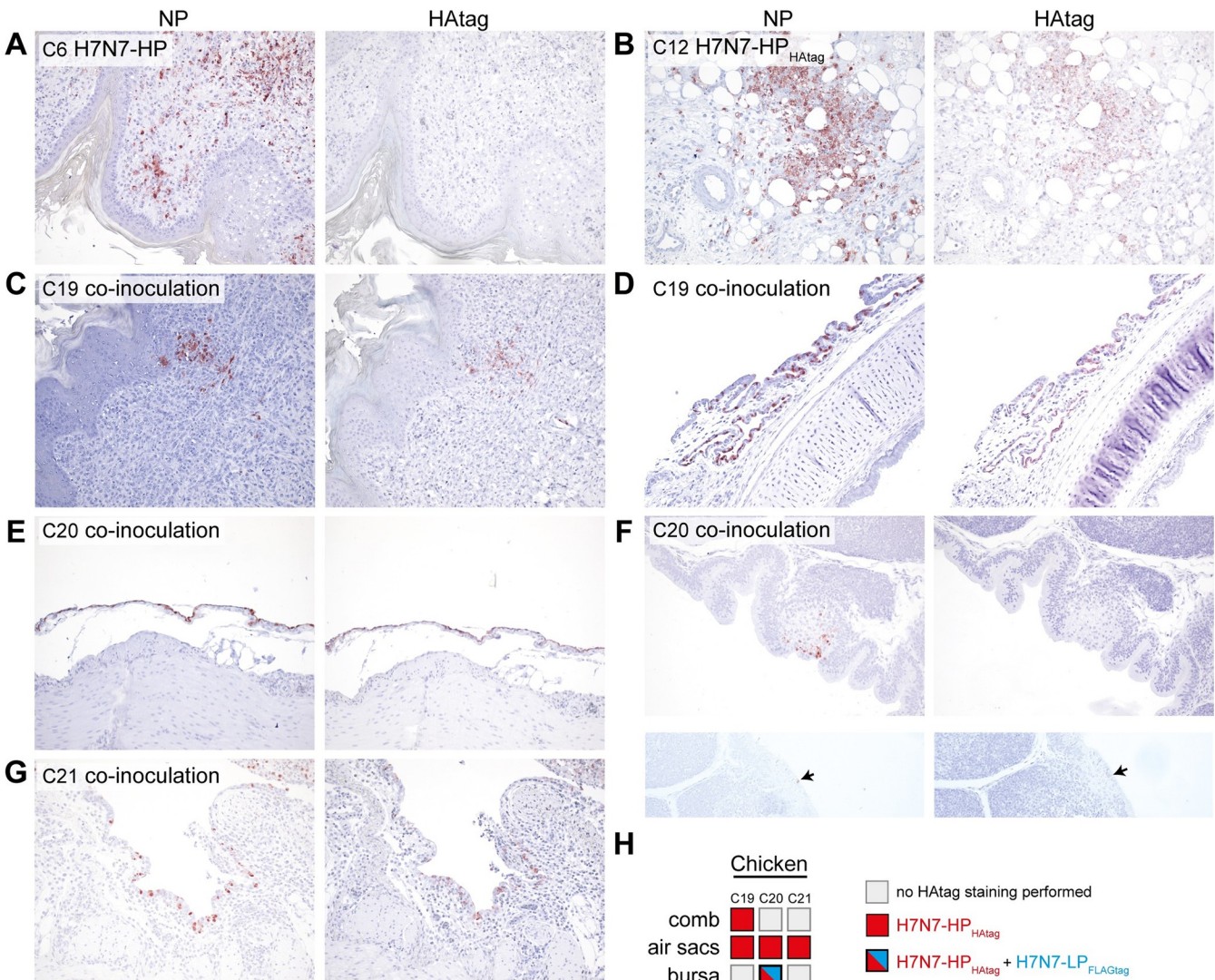

**Fig 7. Detection of H7N7-HP$_{HAtag}$- and H7N7-LP$_{FLAGtag}$-infected cells in tissues of co-inoculated chickens.** Detection of nucleoprotein (NP; left panels) and HAtag (right panels) expression on sequential slides as determined by immunohistochemistry on tissues that scored '2' by NP immunostaining. (A) As negative control for HAtag staining, comb of a H7N7-HP mono-inoculated chicken, harvested at 4 dpi. (B) As positive control for HAtag staining, comb of a H7N7-HP$_{HAtag}$ mono-inoculated chicken, harvested at 3 dpi. (C) Comb of a co-inoculated chicken, which was harvested at 3 dpi. (D) Air sacs surrounding the bronchus of a co-inoculated chicken, which was harvested at 3 dpi and tested positive for HPAIV RNA and PFU. (E) Air sacs surrounding the oesophagus of a co-inoculated chicken, which was harvested at 3 dpi and tested positive for HPAIV RNA and PFU. (F) Two areas of bursa of a co-inoculated chicken, which was harvested at 3 dpi and tested positive for both HPAIV and LPAIV RNA and PFU. The arrowheads indicate NP- and HAtag-positive area. (G) Air sacs surrounding the oesophagus of a co-inoculated chicken, which was harvested at 3 dpi and tested positive for HPAIV RNA and PFU. (H) Overview of the virus detected in the co-inoculated chickens from panels C-G. Magnification 200x.

## Discussion

Although LPAIVs primarily circulate in wild aquatic birds, the emergence of newly converted HPAIVs is associated with terrestrial poultry. We hypothesized that HPAIVs have a selective advantage over LPAIVs in chickens, which would be absent or less strong in ducks, because pathogenesis and viral tropism of LPAIVs and HPAIVs differ between wild aquatic birds and terrestrial poultry. Especially systemic dissemination and replication potency of HPAIVs differ markedly between these bird species. To test this hypothesis, the intrahost selection of a minority of HPAIVs from a majority of LPAIVs was monitored in co-inoculated chickens and

ducks. HPAIV was selected in four of six chickens. In contrast, co-inoculated ducks predominantly shed LPAIV, despite detection of infectious HPAIV in tissues at early time points. Of note, the experiments performed within this study were designed to provide a qualitative assessment of species-specific intrahost HPAIV selection. A limitation of such an approach is the small number of animals that can be analyzed in an in-depth manner, especially when investigating large-sized animals. Nevertheless, despite the interindividual variation observed in the co-inoculated animals, the patterns of virus selection were clearly distinct between the two species. From these experiments we can conclude that the H7N7 HPAIV tested here harbors a selective advantage over H7N7 LPAIV in chickens but not in ducks. These intrahost selection differences might explain the species-restricted nature of HPAIV emergence.

Intrahost selection of a minor variant depends on its intrinsic selective advantage, antagonism enacted on the minor by the major variant, and timing, and has a stochastic component. The observation that H7N7-HP$_{HAtag}$ viral antigen expression and infectious virus titers were lower in co-inoculated than mono-inoculated chickens implies that H7N7-LP$_{FLAGtag}$ partially antagonizes H7N7-HP$_{HAtag}$ replication. Still, the selective advantage of H7N7-HP$_{HAtag}$ over H7N7-LP$_{FLAGtag}$ was sufficient for HPAIV to be selected in four of six chickens inoculated at a 1:62 ratio. Similar observations were reported by Graaf et al., who showed antagonism of LPAIV on HPAIV replication, but also that HPAIV was selected at HPAIV/LPAIV inoculation ratios exceeding 1:100, using a natural H7N7 pair of LPAIV/HPAIV from Germany [35]. In contrast, Beerens et al. observed selection of H7N3 HPAIV in all chickens co-inoculated with HPAIV/LPAIV at a far lower ratio (of 1:1674, as determined based on next-generation sequencing read counts) [13]. A trend towards LPAIV potentiating rather than antagonizing HPAIV pathogenesis and selection in chickens was observed by Bessière et al. [36]. Varying outcomes of the experimental co-inoculation studies point at a high strain-dependency. For example, the divergent results of Bessière et al. might be because they used a duck Gs/Gd-lineage H5N8 HPAIV, which is highly pathogenic for ducks and therefore not representative of newly converted HPAIVs. It should also be noted that HPAIV selection within experimental co-inoculation studies might be facilitated by (i) the high HPAIV/LPAIV ratios, overestimating the HPAIV frequency in the initial phase of their emergence, (ii) the concomitant inoculation of LPAIV/HPAIV, mimicking HPAIV genesis early during LPAIV infection, and by (iii) the route of inoculation.

The selection of H7N7-HP$_{HAtag}$ in co-inoculated chickens might have been facilitated by the fact that H7N7-LP$_{FLAGtag}$ had characteristics of a wild bird-adapted virus. Generally, chicken-adapted viruses replicate well in the respiratory and enteric tracts of chickens, whereas wild bird-adapted viruses do not infect chickens efficiently [37–40]. Here, we showed OP H7N7-LP$_{FLAGtag}$ shedding but limited virus detection in tissues of mono-inoculated chickens, whereas H7N7-LP$_{FLAGtag}$ replicated well in ducks. Antagonism conferred by a well-replicating chicken-adapted LPAIV on HPAIV replication and ensuing selection might be stronger than that conferred by a wild bird-adapted LPAIV. This implies that HPAIV natural selection might be favored directly following the event of LPAIV spillover from wild birds to poultry rather than during circulation of a poultry-adapted LPAIV. The time of LPAIV circulation in poultry before HPAIV emergence ranged from days to months or years [41], which might have then been influenced by different degrees of poultry-adaptation of the progenitor virus. However, further experiments assessing selection efficacy of HPAIV/LPAIV pairs with varying degrees of poultry-adaptation phenotypes are necessary to prove or disprove this hypothesis.

The location of HPAIV intrahost selection, in the same or separate compartment than where the LPAIV replicates, is yet unknown. Beerens et al. reported on the simultaneous detection of HPAIV and LPAIV in a flock of turkeys showing signs of respiratory disease, in which the highest frequency of HPAIV genome copies (4.4%) was detected in the lungs [13].

However, as the turkeys were slaughtered before the HPAIV could outcompete the LPAIV, it is unknown whether or in which site HPAIV intrahost selection would have occurred. Here, the replication of H7N7-HP$_{HAtag}$ in air sac epithelial cells of co-inoculated chickens at 3 dpi was striking, as it was present in all three co-inoculated chickens whereas it was absent in mono-inoculated chickens. Concomitantly, systemic dissemination had already occurred and therefore we propose that HPAIV selection might have happened in the separate compartment of the air sacs, where H7N7-LP$_{FLAGtag}$ was not detected, but was promoted by systemic replication. Seekings et al. also reported that MBCS-containing AIVs were preferentially selected in sites within embryonated chicken eggs where LPAIV was not replicating [42]. Of note, endothelial cells in the air sacs of co-inoculated chickens were NP-negative, indicating that if dissemination to the cardiovascular system happened in the air sacs, it was not accompanied by endothelial cell infection.

H7N7-HP$_{HAtag}$ tropism, shedding and replication levels were reduced in mono-inoculated ducks compared to chickens, which is in accordance with results from previous comparative studies on non-Gs/Gd HPAIVs in both species [24,26,27,32,43]. Furthermore, despite the detection of H7N7-HP$_{HAtag}$ RNA in OP and CL swabs from ducks, infectious virus was only recovered from the latter. Comparing these results to previous published studies on HPAIV shedding from ducks is challenging because excretion of infectious virus, next to RNA-based analyses, is rarely investigated [24,25,27,43–47]. Whether the detection of H7N7-LP$_{FLAGtag}$ and H7N7-HP$_{HAtag}$ in respectively the digestive and respiratory tracts of mono-inoculated ducks can be attributed to intrinsic virus traits or to altered infection kinetics cannot be determined due to the ~100-fold difference in inoculation dose. In literature, replication in both the respiratory and intestinal tracts has been reported upon experimental inoculation of ducks with LPAIVs and non-Gs/Gd HPAIVs [23–27,43,48].

Strikingly, RNA and low levels of infectious virus of H7N7-LP$_{FLAGtag}$ and H7N7-HP$_{HAtag}$ were detected in tissues outside the respiratory and digestive tracts in mono- and/or co-inoculated ducks suggesting their systemic dissemination. Widespread detection of non-Gs/Gd HPAIV RNA and infectious virus in duck tissues has been reported before [24–27], without virus isolation from blood [27], but is very rare for LPAIVs in ducks [24–26,32,38]. Here, we have found no convincing evidence of active LPAIV or HPAIV replication, either by detection of viral antigen by IHC or genomic viral RNA by ISH, outside of the respiratory and digestive tracts of ducks. The widespread detection of (primarily H7N7-HP$_{HAtag}$) viral RNA in homogenized tissues might be a result of abortive infection or replication below the limit of detection of analyses performed on fixed tissues. The co-sampling of virus-positive air sacs might have cross-contaminated some internal organs. Why HPAIVs do not efficiently disseminate systemically nor replicate in the endothelium of wild aquatic birds has not been elucidated [31]. Species-specific endotheliotropism might be a result of differences in antiviral immune responses and/or immune response regulation within endothelial cells [49,50]. Importantly, chickens lack the pattern recognition receptor RIG-I and therefore sense influenza virus RNA differently than other species, for instance through MDA5 [51–54].

We are the first to show, using a HPAIV/LPAIV virus pair that represents natural LPAIV to HPAIV conversions, that, as hypothesized, co-inoculation of ducks with H7N7-HP$_{HAtag}$ and H7N7-LP$_{FLAGtag}$ results in the predominance of the latter, which was primarily evident from the shedding data. In contrast, Bessière et al. reported shedding of HPAIV RNA in all ducks co-inoculated with HPAIV/LPAIV at a 1:1000 ratio using a virus pair based on H5N8 Gs/Gd-HPAIV [36]. The difference in results might be explained by the facts that (i) the H5N8 LPAIV did not result in an expected shedding pattern of a wild bird LPAIV in ducks [48], i.e., prolonged OP shedding and high CL shedding, and that (ii) H5N8 Gs/Gd-HPAIV is highly pathogenic in ducks, which is not representative of newly converted HPAIVs [24,25]. Here, the

predominance of H7N7-LP$_{FLAGtag}$ in co-inoculated ducks was maintained despite early replication of H7N7-HP$_{HAtag}$ in tissues including the air sacs. H7N7-HP$_{HAtag}$ infectious virus titers in duck tissues were low and systemic replication of H7N7-HP$_{HAtag}$ was inefficient, like in the H7N7-HP$_{HAtag}$ mono-inoculated ducks. Therefore, H7N7-HP$_{HAtag}$ replication was not sufficient to result in infectious virus shedding and only H7N7-LP$_{FLAGtag}$ was shed from the OP and CL routes.

The process of HPAIV emergence can be divided into (i) HPAIV genesis, (ii) HPAIV intrahost selection, and (iii) HPAIV interhost transmission. Here, we provide evidence that HPAIV intrahost selection is species-specific, but we did not investigate species-specific HPAIV genesis nor population-level transmission. There is evidence that MBCS acquisition, i.e., HPAIV genesis, can probably occur in non-terrestrial poultry hosts, exemplified by the rare detection of MBCS-containing AIVs in wild birds [55,56] and the generation of HPAIVs in mammalian laboratory systems [57]. In the absence of contact-exposed animals, we measured RNA and infectious virus excretion by collecting OP and CL swabs and by sampling drinking water as a proxy for the likelihood of interhost transmission. The detection of both viruses in the water from co-inoculated chickens and only H7N7-LP$_{FLAGtag}$ in the water from co-inoculated ducks implies that the likelihood of H7N7-HP$_{HAtag}$ transmission among ducks was low. However, transmission rates among birds in natural settings are influenced by many factors that are not well recapitulated in the current experimental setup. Large differences exist between domestic and wild bird ecosystems with regard to population dynamics and environment. It has been theorized that natural ecosystems of wild aquatic birds select for low-virulent pathogens as highly virulent pathogens, leading to increased predatory risk and reduced migratory capacities, would not be selected [58]. Adversely, immunologically naïve and uniform predator-free domestic bird populations consisting of juveniles at high densities would allow the persistence of virulent viruses. However, the theory assumes that HPAIVs are inherently pathogenic for wild aquatic birds, which is not the case for non-Gs/Gd MBCS-containing viruses. Furthermore, the circulation of virulent Gs/Gd-H5 HPAIVs in wild birds shows that natural ecosystems can support transmission of virulent viruses.

Taken together, we have shown that the intrahost selection of a minority of H7N7 HPAIVs from a majority of LPAIVs is species-dependent. HPAIV harbored a higher selective advantage over LPAIV in chickens than in ducks, likely reflecting the large difference in efficient systemic HPAIV replication between the two species. Species-specific intrahost selection likely contributes, amongst other host and environmental factors, to the species-restriction that is associated with the emergence of newly converted HPAIVs.

## Material and methods

### Cell culture

Chicken and duck embryonic fibroblasts (CEF; DEF) were isolated from 11-day-old chicken (Drost, Loosdrecht, The Netherlands) and 13-day-old duck embryos (Duck-To-Farm B.V., Ermelo, The Netherlands) (protocol adapted from [59]) and cultured in Medium 199 (LONZA) supplemented with 10% Fetal Calf Serum (FCS; Greiner Bio-One), 10% Tryptose Phosphate Broth (TPB; MP Biomedicals), 100 U/mL penicillin (LONZA), and 100 U/mL streptomycin (LONZA). Madin–Darby canine kidney cells (MDCK) were cultured in Eagle's Minimal Essential Medium (LONZA) supplemented with 10% FCS, 100 U/mL penicillin, 100 U/mL streptomycin, 2 mM L-glutamine (LONZA), 1.5 mg/mL sodium bicarbonate (LONZA), 20 mM HEPES (LONZA), and non-essential amino acids (NEAA; LONZA). Human Embryonic Kidney-293T (HEK-293T) cells and DF-1 chicken fibroblasts were cultured in Dulbecco's Modified Eagle Medium (DMEM; LONZA) supplemented with 10% FCS, 100 U/mL

penicillin, 100 U/mL streptomycin, 2 mM L-glutamine, 1 mM sodium pyruvate (Gibco), and NEAA. HEK-293T were maintained in the presence of 0.5 mg/mL Geneticin (Invitrogen). Cells of mammalian origin were cultured at 37°C in 5% $CO_2$ and avian cells at 39°C in 5% $CO_2$.

## Virus production and propagation

A/mallard/Netherlands/12/00 H7N3 (H7N3-LP) was generated by reverse genetics in HEK-293T cells as described previously [60] and propagated twice in MDCK and once in embryonated chicken eggs (ECE). Epitope tags, linker sequence, and MBCS alterations were introduced in the bidirectional pHW2000 plasmid encoding HA of H7N7 A/chicken/Netherlands/1/03 by site-directed mutagenesis as previously described [61]. All A/chicken/Netherlands/1/03 H7N7 recombinant viruses were generated by reverse genetics and propagated twice in ECE with the exception of H7N7-LP, which was propagated twice in MDCK cells and once in ECE. The virus stocks were sequenced using Sanger sequencing to confirm that the coding sequences were identical to the GISAID strain entries. Experiments with LPAIVs were performed under biosafety level 3 (BSL3) or BSL3+ conditions and experiments with HPAIVs were performed under BSL3+ conditions.

## Virus titration

Virus titers were determined by endpoint titration in MDCK cells (virus stocks, replication kinetics supernatant, swabs, homogenized tissue samples) or ECE (virus stocks, inoculum). The sensitivities of both endpoint titration methods were compared using OP and CL swabs collected from three H7N7-LP$_{HAtag}$-inoculated ducks. Titration in ECE showed an on average 10-fold higher sensitivity than MDCK cells (S12 Fig), yet patterns were similar. Due to practical considerations, the bulk of the titrations were performed in MDCK cells. Briefly, MDCK cells or ECE were inoculated with 100 µL of tenfold serial dilutions of the sample in infection medium, i.e., MDCK medium with 0.4 µg/mL tosyl-phenylalanyl-chloromethyl-ketone (TPCK)-treated trypsin (Merck) without FCS. MDCK cells were incubated with sample for 1 h, after which the cells were washed once (virus stocks, replication kinetics supernatant) or twice (first three dilutions of swabs and homogenized tissue samples) with infection medium and incubated for 3 days in 200 µL infection medium. Inoculated ECE were incubated for 2 days. MDCK supernatant or ECE allantoic fluid was tested for agglutinating activity using turkey red blood cells as an indicator of virus presence. Infectious virus titers were calculated from 8 replicates (virus stocks) or 4 replicates (replication kinetics supernatant, inoculum, swabs, and homogenized tissues) by the method of Spearman-Karber and expressed as median tissue culture infectious dose per mL (TCID$_{50}$/mL) or median egg infectious dose per mL (EID$_{50}$/mL). Titers of homogenized tissue samples were normalized per gram tissue (TCID$_{50}$/gram). The limit of detection for the weight-normalized samples was calculated from the heaviest sample per tissue per experimental group.

## Flow cytometry

DF-1 monolayers in 6-well dishes at 90% confluence were inoculated with H7N7-LP$_{FLAGtag}$ or H7N7-HP$_{HAtag}$ at an MOI of 0.1 (TCID$_{50}$). After 1 h of incubation, cells were washed thrice with phosphate buffered saline (PBS) and overlaid with DMEM containing 0.08 µg/mL TPCK-treated trypsin without FCS. At 24 hpi, cells were trypsinized, and fixed and permeabilized with BD Cytofix/Cytoperm (BD Biosciences) according to manufacturer's instructions. Cells were subsequently stained for 1 h on ice with the following antibodies diluted in BD Perm/Wash buffer: 20 µg/mL anti-influenza NP antibody (H16-L10-4R5 (ATCC HB-65)), followed

by 10 μg/mL Alexa Fluor488-conjugated goat anti-mouse IgG2α antibody (A-21131; Thermo Fisher Scientific); 20 μg/mL anti-FLAGtag antibody (M2; Sigma Aldrich), followed by 10 μg/mL Alexa Fluor488-conjugated goat anti-mouse IgG antibody (A11029; Thermo Fisher Scientific); 7 μg/mL FITC-conjugated anti-HAtag antibody (H7411; Sigma Aldrich). After each staining, cells were washed twice with BD Perm/Wash buffer. Cells were subjected to flow cytometry on the FACSLyric (BD Biosciences) and the data were analyzed using FlowJo v10.7.2 software (BD Biosciences).

## Replication kinetics

Monolayers of CEF and DEF in 6-well dishes at 90% confluence were inoculated in duplicate with tagged and wild-type H7N7 viruses at an MOI of 0.001 ($TCID_{50}$). After 1 h of incubation, the cells were washed thrice with PBS and 4 mL Medium 199 without FCS and TPB was added. LPAIV- and HPAIV-inoculated cells were incubated in the presence and absence of 0.08 μg/mL TPCK-treated trypsin respectively. Cells were incubated at 39°C in 5% $CO_2$ and supernatant was harvested at the indicated time points and stored at -80°C until further processing. Viral titers were determined by endpoint titration in MDCK cells in the presence of 0.4 μg/mL TPCK-treated trypsin as described above.

## Animal experiments

**Ethics statement.** Animal experiments were conducted according to the European guidelines (EU directive on animal testing 86/609/EEC) and Dutch legislation (Experiments on Animals Act, 1997). The experimental protocols (permit number AVD1010020186744; protocols 18-6744-02, 18-6744-05, 18-6744-06, 18-6744-07, 18-6744-08) were reviewed and approved by an ethical committee at the Erasmus Medical Center, Rotterdam, The Netherlands. Ethical review was waived for the procedures involving embryonated eggs as embryos (regardless of gestational stage) from avian species are not subjected to ethical regulations in the European Union. Please see the "European directive on the protection of animals used for scientific purposes" (Article 1, point 3). https://eur-lex.europa.eu/legal-content/EN/TXT/HTML/?uri=CELEX:32010L0063&from=EN (accessed on the 14th of October 2022).

**Animals.** Specific pathogen-free female 6-week-old White Leghorn chickens (*Gallus gallus domesticus*) were obtained from Royal GD (Deventer, The Netherlands). One-day-old mixed-sex domestic ducks (*Anas platyrhynchos domesticus*) were obtained from Duck-To-Farm B.V. (Ermelo, The Netherlands) and raised in captivity until 6-weeks-old at the animal care facility at the ErasmusMC, Rotterdam, The Netherlands. Blood samples were collected 1 week prior to inoculation to confirm H7 seronegativity by hemagglutination inhibition assay [62].

**Inoculation doses.** The percentage of HPAIV in a pool of LPAIV was set at 1% HPAIV ($10^4$ $EID_{50}$) and 99% LPAIV ($10^6$ $EID_{50}$), and therefore those same inoculation doses were used during all mono-inoculation experiments, i.e., $10^4$ $EID_{50}$ for all HPAIV inoculations and $10^6$ $EID_{50}$ for all LPAIV inoculations. Actual inoculation dose was determined by endpoint titration in ECE (S1B Fig). Based on the back-titration, the percentages of HPAIV/LPAIV in the co-inoculation experiment were 2% H7N7-HP$_{HAtag}$ ($10^{4.3}$ $EID_{50}$) and 98% H7N7-LP$_{FLAGtag}$ ($10^{6.1}$ $EID_{50}$), i.e., a 1:62 ratio.

**Animal experimental design.** All animal experimental groups were divided over 5 separate experiments (S1B Fig). Birds were housed per 6 animals in class III isolators within negatively pressurized BSL3+ containment facilities and were allowed to acclimatize for 3 days. They were subsequently inoculated intratracheally and intra-oesophageally with 200 μL of virus inoculum per route for chickens and 1.5 mL per route for ducks. Body weight was recorded daily and birds were scored for disease severity thrice daily (S1A Fig). Disease

severity scores were defined as follows: 0, normal; 1, drowsiness, less active, hunched posture; 2, edema of the head; 3, lethargic and start of hemorrhaging on the head or feet; 4, severe hemorrhaging on the head or feet, only active upon strong stimulation; 5, no activity upon stimulation. Oropharyngeal, cloacal, and drinking water swabs were collected daily in 1 mL of virus transport medium (VTM; Eagle's-MEM with Hank's balanced salt solution (Thermo Fisher Scientific), 25 mM HEPES, 10% glycerol, 100 U/mL polymyxin B sulphate (Sigma-Aldrich), 0.5% lactalbumin hydrolysate (Sigma-Aldrich), 100 U/mL nystatin (Sigma-Aldrich), 50 mg/mL gentamicin (Gibco), 100 U/mL penicillin, and 100 μg/mL streptomycin) and stored at -80˚C for subsequent RNA isolation for RT-qPCR and endpoint titration. On 3 and 7 dpi, 3 birds were euthanized by exsanguination under anesthesia and 21 tissues were harvested for immunohistochemical and/or virological analysis. For the former, tissues were placed in 10% neutral buffered formalin and for the latter tissues were placed in 3 mL of VTM, homogenized using ceramic lysing spheres (MP Biomedicals), and stored at -80˚C for RNA isolation for RT-qPCR and endpoint titration. During dissections, utensils were cleaned with chlorine and 70% ethanol solutions to minimize contamination of both infectious virus and viral RNA between tissue samples. Additional euthanasia and dissections were performed when one of the humane endpoint criteria was met: severe respiratory distress, severe neurological symptoms, or no activity upon stimulation.

**Detection of viral RNA in swabs and tissues.** Total nucleic acid isolations were performed using an in-house developed method, as previously described [63]. Briefly, 100 μL of swab sample was added to 150 μL MagNaPure 96 external lysis buffer (Roche). Alternatively, 200 μL homogenized tissue sample was added to 300 μL lysis buffer. 150 μL of the resulting mixture was added to 50 μL AMPureXP magnetic beads (Beckman Coulter) and incubated for 15 min at room temperature. The plate was placed on a magnetic block (DynaMag–96 Side Skirted Magnet; Thermo Fisher Scientific) to allow for the beads with bound nucleic acids to displace towards the magnet. The beads were washed thrice with 70% ethanol, air-dried for 1 min, and eluted in 50 μL PCR-grade water. For RT-qPCR, 5 μL of RNA was added to a mix containing 1 μL primer/probe mix targeting the influenza virus matrix segment (forward primer CTTCTRACCGAGGTCGAAACGTA; reverse primer TCTTGTCTTTAGCCAYTC CATGAG; probe 6-FAM TCAGGCCCCCTCAAAGCCGARA BHQ1) or 0.67 μL of each primer and probe targeting the H7 cleavage site for LPAIV/HPAIV discrimination (forward primer GGCAACAGGAATGAAGAATGTTC 40 μM; reverse primer CTTCCCATCCATT TTCAATGAAAC 40 μM; LPAIV probe 6-FAM CACCAAATAGGCCTCTTCCCTTTGGGA BHQ1 10 μM; HPAIV probe Dragonfly CAAATAGGCCTCTCCTCCTCCTCTTTGGG BHQ2 10 μM), 4X TaqMan Fast Virus 1-Step Master Mix (Thermo Fisher Scientific) and PCR-grade water for a total of 20 μL. The following cycling program was used on an Applied Biosystems ABI7500 Real-Time PCR system (Thermo Fisher Scientific): 5 min 50˚C, 20 s 95˚C, (3 s 95˚C, 31 s 60˚C) × 45 cycles. Viral RNA levels are depicted as 40-cycle threshold value (40-Ct). The sensitivity of the LPAIV/HPAIV-discriminating primer/probe set was tested with a range of LPAIV/HPAIV virus mixtures at different ratios. For the comparison of viral RNA levels in tissues, 40-Ct values were normalized to 1 gram: $(40\text{-}Ct)+\log_2(1/\text{tissue weight (in gram)})$. For example, a tissue weighing 0.5 gram and harboring 40-Ct = 20 had a normalized 40-Ct value of 21 $((20)+\log_2(1/0.5) = 21)$.

**Plaque assay.** Swabs and tissue samples, harvested at 3 dpi, of co-inoculated birds were subjected to an LPAIV/HPAIV discriminating plaque assay. MDCK monolayers at 95% confluence were inoculated with 500 μL of EMEM infection medium with TPCK-treated trypsin containing a serial dilution of swab or homogenized tissue sample. Virus stocks of H7N7-HP$_{HAtag}$ and H7N7-LP$_{FLAGtag}$ were included as controls. After 1 h of incubation at 37˚C, cells were washed once with PBS and a 2 mL overlay containing a 1:1 ratio of 2x EMEM

(Capricorn Sciences) and 2.4% Avicel (FMC BioPolymer) in $H_2O$ with 1.6 μg/mL TPCK-treated trypsin was overlaid. At 28 hpi, the cells were washed twice with PBS and fixed overnight at -20°C in 500 μL 80% acetone. The plates were submerged in 70% ethanol to allow removal from the BSL3+ lab. The plates were washed with PBS, and stained overnight at 4°C with a mixture of primary antibodies diluted in PBS with 0.1% bovine serum albumin (BSA): 1 μg/mL anti-influenza virus NP antibody (mouse IgG2α; ATCC HB-65) and 44 ng/mL anti-HAtag antibody (rabbit IgG; C29F4; Cell Signaling). FLAGtag staining of the plaques was not performed due to aspecific background staining of plaques when using anti-FLAG antibodies. Plates were washed thrice with PBS and incubated with a mixture of secondary antibodies in PBS with 0.1% BSA overnight at 4°C: 4 μg/mL AF488-conjugated goat anti-mouse IgG2α (A21131; Thermo Fisher Scientific) and 4 μg/mL AF647-conjugated anti-rabbit IgG (A21244; Thermo Fisher Scientific). Plaques were scanned on the Amersham Typhoon (GE Healthcare) in the Cy2 and Cy5 channels at 50 μm resolution and 300 PMT voltage. Images from the 2 channels were adjusted for contrast and compiled using FIJI software (National Institutes of Health). Plaques positive for solely NP staining (AF488) were counted as LPAIV and plaques positive for both NP and HAtag (AF647) were counted as HPAIV. The infectious virus titers were expressed as plaque forming units per mL (PFU/mL) or were normalized per gram tissue (PFU/gram).

**Immunohistochemistry.** After 2 weeks of fixation in 10% neutral buffered formalin, tissues were embedded in paraffin. Nasal conchae, trachea, and bronchus were decalcified for 4 days in 10% EDTA prior to paraffinization. Thin (3 μm) sequential sections were prepared for immunohistochemical (IHC) analysis and a hematoxylin and eosin staining. Formalin-fixed paraffin-embedded (FFPE) sections were deparaffinized with xylene and rehydrated using graded alcohols, and antigens were retrieved by 0.1% protease XIV from Streptomyces griseus (Sigma-Aldrich) treatment in PBS for 10 min at 37°C for the influenza NP staining or by boiling for 15 min in 10 mM citric acid (pH = 6) for HAtag staining. Endogenous peroxidase activity was blocked by treatment with 3% $H_2O_2$ in PBS for 10 min at room temperature. For the influenza virus NP staining, sections were incubated with 5 μg/mL anti-NP antibody (ATCC HB-65), or mouse IgG2α isotype control (MAB003; R&D Systems) for 1 h at room temperature in PBS/0.1% BSA, followed by 1 h incubation with 10 μg/mL horse radish peroxidase (HRP)-conjugated goat-anti-mouse IgG2α (Star133P; Bio-Rad). For HAtag staining, sections were incubated with 0.66 μg/mL HAtag antibody (rabbit IgG; C29F4), or rabbit IgG isotype control (AB-105-C; R&D systems) overnight at 4°C, followed by 1 h incubation with 1:200 HRP-conjugated goat-anti-rabbit IgG (P021702-2; Agilent). FLAGtag staining on FFPE tissues was not performed due to aspecific background staining when using anti-FLAG antibodies. Peroxidase activity was revealed using 3-amino-9-ethyl-carbazole (Sigma-Aldrich) in N,N-dimethylformamide (Honeywell Fluka) diluted in a final concentration of 0.0475 M of NaAc (pH = 5) with 0.05% of $H_2O_2$ for 10 min at room temperature, resulting in a bright red precipitate. Sections were counterstained with hematoxylin and mounted with Kaiser's glycerol gelatin (VWR). Lung sections from a ferret infected with pandemic H1N1 were included in each NP staining as positive control. Micrographs were taken on the Axio Imager.A2 using ZEN software (Zeiss) and white balance and contrast were adjusted using Adobe Photoshop 2022. The degree of viral antigen expression was based on the presence of NP in nuclei and scored as negative (0), sporadic single positive cells (1), multiple sites of positive cells (2), and diffuse positivity (3).

**Influenza A virus In-Situ Hybridization.** RNAScope RNA probes were designed targeting conserved parts of the Influenza A virus matrix and nucleoprotein vRNA (Bio-Techne Ltd; V-Influenza-H1N1_H5N1-M-NP). In-situ hybridization was performed on FFPE consecutive sections from the indicated chicken and duck tissues using RNAScope 2.5 HD Reagent Kit-

RED (Bio-Techne Ltd), as described by the manufacturer. Influenza A virus matrix and nucleoprotein vRNA molecules were visualized as red chromogenic dots. Positive (Ubiquitin C; UBC) and negative (Bacillus subtilis strain SMY dihydrodipicolinate reductase; DapB) control probes were included for each tissue. Lung sections from a ferret infected with pandemic H1N1 were included as positive control for influenza A virus infection.

**Statistics.** Statistical analyses were performed using Graphpad version 9.2.0.

## Supporting information

**S1 Fig. Overview of experimental design and division of animal groups over five experiments.** (A) Schematic overview of the experimental design of the in vivo experiments. Six six-week-old chickens or ducks were inoculated intratracheally and intra-oesophageally with H7 avian influenza viruses. Disease severity was monitored thrice daily and body weight measurements, oropharyngeal (OP), cloacal (CL), and drinking water swabs were collected daily. Three animals were euthanized at three dpi and at seven dpi, or at earlier time points when animals reached humane endpoints, to harvest tissues for virological and histological analysis. (B) Overview of chicken and duck mono- and co-inoculation groups as divided over five experiments, indicating species, animal identifiers, virus, intended inoculation dose (aim), and actual inoculation dose as determined by endpoint titration of the inoculum in embryonated chicken eggs and expressed as $\log_{10}$ EID$_{50}$.
(TIF)

**S2 Fig. Changes in body weight upon inoculation of chickens and ducks with H7N7 LPAIVs and HPAIVs.** Body weight was recorded daily in mono-inoculated ducks (A) and chickens (B) and co-inoculated chickens and ducks (C) and is presented relative to the body weight on the day of inoculation.
(TIF)

**S3 Fig. Shedding and replication in tissues of wild-type and epitope tagged H7N7 LPAIVs in ducks.** (A-D) Viral RNA quantification in oropharyngeal (OP (top panels)) and cloacal (CL (bottom panels)) swabs of ducks mono-inoculated with ~$10^6$ EID$_{50}$ of H7N3-LP (A), H7N7-LP (B), H7N7-LP$_{FLAGtag}$ (C), or H7N7-LP$_{HAtag}$ (D). Viral RNA amounts were determined by RT-qPCR targeting the influenza virus matrix segment and expressed as 40-cycle threshold (Ct). (E) Viral RNA quantification in tissues harvested at 3 and 7 dpi from ducks inoculated with ~$10^6$ EID$_{50}$ of H7N3-LP, H7N7-LP, H7N7-LP$_{FLAGtag}$, or H7N7-LP$_{HAtag}$. Viral RNA amounts were determined by RT-qPCR targeting the influenza virus matrix segment and expressed as 40-Ct normalized/gram tissue. Bars represent the arithmetic mean and symbols indicate data from individual animals as in (A-D). The dotted lines distinguish tissues from the respiratory, digestive, and miscellaneous systems. The asterisks indicate that data are absent.
(TIF)

**S4 Fig. Presence of virus in drinking water of chickens and ducks mono-inoculated with LPAIVs and HPAIVs.** (A) Viral RNA quantification in swabs taken from the drinking water of ducks mono-inoculated with ~$10^6$ EID$_{50}$ of H7N7-LP, H7N7-LP$_{FLAGtag}$, H7N7-LP$_{HAtag}$, or H7N3-LP, or ~$10^4$ EID$_{50}$ of H7N7-HP$_{HAtag}$. Viral RNA amounts were determined by RT-qPCR targeting the influenza virus matrix segment and expressed as 40-cycle threshold (Ct). (B) Viral RNA quantification in swabs taken from the drinking water of chickens mono-inoculated with ~$10^4$ EID$_{50}$ of H7N7-HP or H7N7-HP$_{HAtag}$, or ~$10^6$ EID$_{50}$ of H7N7-LP$_{FLAGtag}$, and analyzed as in (A). (C, D) Infectious virus titers in swabs taken from the drinking water of ducks (C) and chickens (D) mono-inoculated with ~$10^4$ EID$_{50}$ of H7N7-HP$_{HAtag}$ or ~$10^6$

$EID_{50}$ of H7N7-LP$_{FLAGtag}$. Infectious titers were determined by endpoint titration in MDCK cells and expressed as $\log_{10}$ TCID$_{50}$/mL. The horizontal dotted lines indicate the limit of detection of the endpoint titration assay.
(TIF)

**S5 Fig. Disease severity and virus shedding and replication in chickens inoculated with H7N7-HP or H7N7-HP$_{HAtag}$.** (A) Severity of disease in six chickens inoculated with ~$10^4$ $EID_{50}$ of H7N7-HP (C1-C6) or H7N7-HP$_{HAtag}$ (C7-C12). Disease severity scores (0–5) were determined thrice daily, each box representing one time point. Dotted lines indicate planned time points for euthanasia and dissection. (B) Viral RNA quantification in oropharyngeal (OP) and cloacal (CL) swabs of individual chickens mono-inoculated with H7N7-HP or H7N7-HP$_{HAtag}$. Viral RNA amounts were determined by RT-qPCR targeting the influenza virus matrix segment and expressed as 40-cycle threshold (Ct). Symbols are as in (A). (C) Viral RNA quantification in tissues harvested at 3 dpi, humane endpoint (HE; C4, C5, C6, C10, and C12), or 7 dpi (C11) from chickens inoculated with ~$10^4$ $EID_{50}$ of H7N7-HP or H7N7-HP$_{HAtag}$. Viral RNA amounts were determined by RT-qPCR targeting the influenza virus matrix segment and expressed as 40-Ct normalized/gram tissue. Bars represent the arithmetic mean and symbols indicate data from individual animals as in (A, B). The dotted lines distinguish tissues from the respiratory, digestive, and miscellaneous systems.
(TIF)

**S6 Fig. Virus shedding from individual H7N7-HP$_{HAtag}$ and H7N7-LP$_{FLAGtag}$ mono-inoculated chickens and ducks.** (A, C) Viral RNA quantification in oropharyngeal (OP) and cloacal (CL) swabs of chickens (A) and ducks (C) mono-inoculated with ~$10^4$ $EID_{50}$ of H7N7-HP$_{HAtag}$ or ~$10^6$ $EID_{50}$ of H7N7-LP$_{FLAGtag}$. Viral RNA amounts were determined by RT-qPCR targeting the influenza virus matrix segment and expressed as 40-cycle threshold (Ct). The symbols indicate data from an individual animal. (B, D) Infectious virus titers in the swabs from (A, C). Infectious titers were determined by endpoint titration in MDCK cells and expressed as $\log_{10}$ TCID$_{50}$/mL. Symbols are as in (A, C). The horizontal dotted lines indicate the limit of detection of the endpoint titration assay.
(TIF)

**S7 Fig. Detection of epitope tagged H7N7 LPAIVs and HPAIVs in tissues of mono-inoculated chickens and ducks euthanized at 7 dpi or due to humane endpoints.** (A, B) Viral RNA quantification in tissues harvested at 7 dpi or at humane endpoint from chickens (A) and ducks (B) inoculated with ~$10^6$ $EID_{50}$ H7N7-LP$_{FLAGtag}$ or ~$10^4$ $EID_{50}$ H7N7-HP$_{HAtag}$. Viral RNA amounts were determined by RT-qPCR targeting the influenza virus matrix segment and expressed as 40-cycle threshold (Ct) normalized/gram tissue. Bars represent the arithmetic mean and symbols indicate data from individual animals. The dotted lines distinguish tissues from the respiratory, digestive, and miscellaneous systems. (C, D) Infectious virus titers in the tissues from (A, B). Infectious titers were determined by endpoint titration in MDCK cells and expressed as $\log_{10}$ TCID$_{50}$/gram tissue. Bars represent the arithmetic mean of $\log_{10}$ transformed values. Symbols and dotted lines similar to (A, B). The horizontal solid lines indicate the limit of detection of the endpoint titration assay per tissue.
(TIF)

**S8 Fig. Viral shedding levels in H7N7-LP$_{FLAGtag}$/H7N7-HP$_{HAtag}$ co-inoculated chickens and ducks as determined by endpoint titration assay in MDCK cells.** Infectious virus quantification in oropharyngeal (OP; solid lines) and cloacal (CL; dotted lines) swabs from H7N7-LP$_{FLAGtag}$/H7N7-HP$_{HAtag}$ co-inoculated chickens (A) and ducks (B). Infectious titers were determined by endpoint titration in MDCK cells and expressed as $\log_{10}$ TCID$_{50}$/mL

(viral RNA levels and plaque forming units are depicted in Figs 4 and 5).
(TIF)

**S9 Fig. Presence of virus in drinking water of H7N7-LP$_{FLAGtag}$/H7N7-HP$_{HAtag}$ co-inoculated chickens and ducks.** Viral RNA quantification in swabs taken from the drinking water of H7N7-LP$_{FLAGtag}$/H7N7-HP$_{HAtag}$ co-inoculated chickens (A) and ducks (B). Viral RNA amounts were determined by HPAIV/LPAIV differentiating RT-qPCR targeting the HA cleavage site and expressed as 40-Ct. (C, D) Infectious virus titers in the swabs from (A, B), depicted as log$_{10}$ PFU/mL as measured by the HPAIV/LPAIV differentiating plaque assay.
(TIF)

**S10 Fig. Virus detection in tissues of H7N7-LP$_{FLAGtag}$/H7N7-HP$_{HAtag}$ co-inoculated chickens and ducks.** (A, B) Infectious virus titers in tissues harvested at 3 dpi from chickens (A) and ducks (B) co-inoculated with H7N7-HP$_{HAtag}$ and H7N7-LP$_{FLAGtag}$. Infectious titers were determined by endpoint titration in MDCK cells and expressed as log$_{10}$ TCID$_{50}$/gram tissue. The horizontal solid lines indicate the limit of detection of the endpoint titration assay per tissue. The dotted lines distinguish tissues from the respiratory, digestive, and miscellaneous systems. (C, D) Viral RNA quantification in tissues harvested at humane endpoint or at 7 dpi from chickens (C) and ducks (D) co-inoculated with H7N7-HP$_{HAtag}$ and H7N7-LP$_{FLAGtag}$. Viral RNA amounts were determined by HPAIV/LPAIV differentiating RT-qPCR targeting the HA cleavage site region and expressed as 40-cycle threshold normalized per gram tissue. Dotted lines are similar as in (A, B). (E, F) Infectious virus titers in the tissues from (C, D). Infectious titers were determined as in (A, B). Dotted and horizontal solid lines are similar as in (A, B).
(TIF)

**S11 Fig. Viral antigen expression in tissues of selected chickens that were euthanized at humane endpoint.** (A) Degree of viral antigen expression in tissues of a chicken mono-inoculated with ~10$^4$ EID$_{50}$ H7N7-HP$_{HAtag}$ and euthanized at humane endpoint at 4 dpi (A) and a chicken co-inoculated with H7N7-LP$_{FLAGtag}$/H7N7-HP$_{HAtag}$ and euthanized at humane endpoint at 3 dpi (B). Viral antigen expression was determined by immunohistochemical detection of the influenza nucleoprotein. Tissues showing no positivity were scored as (0), those showing sporadic single positive cells as (1), multiple sites of positive cells as (2), and diffuse positivity as (3).
(TIF)

**S12 Fig. Comparison of infectious virus titers as determined by endpoint titration assay in embryonated chicken eggs or in MDCK cells.** Infectious virus titers in oropharyngeal (OP) and cloacal (CL) swabs, as determined by endpoint titration assay in embryonated chicken eggs or in MDCK cells, of the three ducks inoculated with ~10$^6$ EID$_{50}$ H7N7-LP$_{HAtag}$ that were followed for seven days (D16-D18). Titers are expressed as log$_{10}$ EID$_{50}$/mL or as log$_{10}$ TCID$_{50}$/mL. Data are depicted as arithmetic mean ± SD calculated from log$_{10}$ transformed values. The horizontal dotted lines indicate the limit of detection of the endpoint titration assays.
(TIF)

**S1 Table. Assessment of virus replication outside of the respiratory and digestive tracts of ducks.**
(DOCX)

## Acknowledgments

We thank Vincent Vaes, Ingeborg van Middelkoop, Elwin Verveer, and Vincent Duiverman for their assistance during animal studies. We thank the Erasmus Laboratory Animal Science

Center (EDC) staff for animal care. We thank Peter van Run for his help with processing animal tissues and Edwin Veldhuis Kroeze, Thijs Kuiken, and Valentina Caliendo for their help with the interpretation of viral antigen stainings.

## Author Contributions

**Conceptualization:** Anja C. M. de Bruin, Mathilde Richard.

**Data curation:** Anja C. M. de Bruin.

**Formal analysis:** Anja C. M. de Bruin.

**Funding acquisition:** Ron A. M. Fouchier, Mathilde Richard.

**Investigation:** Anja C. M. de Bruin, Monique I. Spronken, Adinda Kok, Miruna E. Rosu, Dennis de Meulder, Stefan van Nieuwkoop, Pascal Lexmond, Mathis Funk, Lonneke M. Leijten, Theo M. Bestebroer, Sander Herfst, Mathilde Richard.

**Methodology:** Anja C. M. de Bruin, Monique I. Spronken, Mathis Funk, Sander Herfst, Debby van Riel, Ron A. M. Fouchier, Mathilde Richard.

**Project administration:** Mathilde Richard.

**Supervision:** Mathilde Richard.

**Visualization:** Anja C. M. de Bruin.

**Writing – original draft:** Anja C. M. de Bruin, Mathilde Richard.

**Writing – review & editing:** Anja C. M. de Bruin, Monique I. Spronken, Adinda Kok, Miruna E. Rosu, Dennis de Meulder, Stefan van Nieuwkoop, Pascal Lexmond, Mathis Funk, Lonneke M. Leijten, Theo M. Bestebroer, Sander Herfst, Debby van Riel, Ron A. M. Fouchier, Mathilde Richard.

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
