## [Decision Letter · Decision Letter 0]

5 Sep 2023

Dear Dr. Richard,

Thank you very much for submitting your manuscript "Species-specific emergence of highly pathogenic avian influenza virus is driven by intrahost selection differences between chickens and ducks" for consideration at PLOS Pathogens. As with all papers reviewed by the journal, your manuscript was reviewed by members of the editorial board and by several independent reviewers. In light of the reviews (below this email), we would like to invite the resubmission of a significantly-revised version that takes into account the reviewers' comments.

While all reviewers thought this study may be of interest to the broader community, they also suggested additional revisions that would improve the quality of the ms. We concur. We also believe that those revisions may require additional experimentation due to concerns about the low number of chickens and ducks used in the experiments, and significant variation in their experimental setup. Additional experiments would strengthen the findings.

We cannot make any decision about publication until we have seen the revised manuscript and your response to the reviewers' comments. Your revised manuscript is also likely to be sent to reviewers for further evaluation.

Sincerely,

Daniel R. Perez, PhD

Academic Editor

PLOS Pathogens

Benhur Lee

Section Editor

PLOS Pathogens

Kasturi Haldar

Editor-in-Chief

PLOS Pathogens

orcid.org/0000-0001-5065-158X

Michael Malim

Editor-in-Chief

PLOS Pathogens

orcid.org/0000-0002-7699-2064

Dear Dr. Richard et al,

I have read the comments of the reviewers and read the ms myself. The reviewers have recommended "Major Revisions" that I believe will improve the quality of the ms. I also believe that those revisions may require additional experimentation due to concerns about the low number of chickens and ducks used in the experiments, and significant variation in their experimental setup. I think additional experiments would strengthen the findings.

Reviewer's Responses to Questions

**Part I - Summary**

Reviewer #1: Low pathogenic avian influenza viruses (LPAIV) of the H5 and H7 subtypes can emerge as HPAIV through the acquisition of a multibasic cleavage site in hemagglutinin. Interestingly, HPAIV have been shown to only emerge in poultry species but not in wild aquatic birds. In this manuscript by de Bruin et al., the authors propose and test the hypothesis that the emergence of HPAIV in poultry is due to species specific selection advantages and that the absence of HPAIV emergence in wild birds is due to selection disadvantages. The authors test this hypothesis by co-inoculating differentially tagged isogenic HPAIV and LPAIV strains at a 1:62 ratio into chickens and ducks, and assess the levels of HPAIV and LPAIV at different times post infection. In four out of six co-inoculated chickens, HPAIV dominated the infection. In contrast, predominantly LPAIV was detected in co-inoculated ducks. Based on these results, the authors conclude that the selection of HPAIV is favored over LPAIV in chickens whereas HPAIV is disfavored in ducks. Overall, this is a very detailed study that provides insight into the emergence of HPAIV only in poultry but not in wild aquatic birds. The authors should address the following concerns.

Reviewer #2: The article titled “Species-specific emergence of highly pathogenic avian influenza virus is driven by intra-host selection differences between chickens and ducks” is a comprehensive experimental study aimed at analyzing the species-specific differences in the selection of different pathotype of H7 avian influenza virus using a mixture of the two at a 1:62 dilution in chickens and ducks. The authors were able to trace LP- and HP-AIV pathotypes by engineering specific tags on the HA. The manuscript is well written, and the research rationale is sound. There are only a few details that need clarification.

Reviewer #3: Strengths:

- The paper examines an important topic: the selection of highly pathogenic influenza viruses within natural hosts. This is particularly relevant as HPAIVs currently pose a tangible threat to animals and humans.

- The hypothesis tested is clear and the results support it.

- The system used is of utmost biological relevance (the results presented are from in vivo experiments using natural hosts).

Originality: I am not an expert on avian influenza and thus I am not sure if the results presented are original, or if they constitute incremental knowledge. Based on the discussion, it seems that the latter is the case.

Weaknesses:

- The main weakness is that the host response to infection by LP or HP or coinfections, was not studied. This is a shame because i) it would likely provide very important information about the processes that drive the observed differences in tropism, shedding and selection; and ii) because the authors have the materials and expertise to tackle this issue easily (see major comments below).

**Part II – Major Issues: Key Experiments Required for Acceptance**

Reviewer #1: Major concerns:

1. Even in the mono-inoculated H7N7-HP ducks, the authors barely detect any infectious virus by TCID50 assay. Is it possible that the infectious dose of HP needed for productive infection in ducks is higher that the dose needed for chickens? These results would be convincing if the authors could perform H7N7-HP (10^6) infections in ducks at a higher dose and establish productive infections.

2. While I agree with the overall conclusions, additional co-inoculation studies in chickens and ducks will strengthen their findings. These experiments were done once with an n=3 for each time point. On Day 7, only 1 out of 3 chickens have high HPAIV titers. Similarly, on day 3, 1 out of 3 chickens have detectable LPAIV. As there is a lot of variation between birds, additional experiments will help rule out any stochastic effects.

3. The organization of supplementary figures in the manuscript needs to be changed. In the current manuscript, body weight, viral RNA, viral titers are organized in separate supplementary figures. While it is easy to compare these parameters across different viruses, it is really hard to follow the results section when panels from different supplementary figures are recalled.

Reviewer #2: 1) Could the author expand on the rationale behind the 1:62 mixture (prior literature)?

2) In both mono- and co-infections, the authors observed widespread and elevated HPAI virus tissue distribution in all infected birds compared to the limited tissue distribution to only gastrointestinal tract of ducks with LPAIV (compatible with previous literature). However, when looking at virus shedding values, it seems that LPAIV is shed OP and CL swabs to higher extent than HPAIV in both chickens and ducks which is counter intuitive. Could the authors comment on the opposite results obtained from tissues compared to swabs?

3) Beside the MBCS, did the H7 virus genome contained any other virulence marker or markers of mammalian adaptation within other genes?

4) Fig 7 recapitulates the tissue distribution of virus antigen as represented by nucleoprotein and HAtag. In subpanel F (C20 co-inoculation), the figure shows positivity for nucleoprotein and not the HA-tag, but in the summary panel H, both HA-tags and FLAG-tag are marked positive. Could the authors demonstrate or comment on the interpretation since as it is displayed seems contradictory?

5) The authors analyze the differential replication of tagged and untagged HP and LP viruses in vitro and in chickens. Is there a reason why the authors did not analyze mono-untagged H7 HPAIV replication in ducks? Differential replication in vivo would be important to further support the authors conclusions on competitive advantage of tagged LP and HP in ducks.

Reviewer #3: The following experiments are not absolutely required. They would make this study much more comprehensive and would provide insights that the study as it stands does not provide.

The authors should examine the host response to infection to the viruses (in single and coinfections) as this will likely provide important information about the differences observed in tropism, shedding and selection. This could be done by using the FFPE tissues that had been collected, and use IHC or IF to quantify markers of (for example) IFN response, inflammation, or any other response that they think could contribute to the observed differences. If antibodies are not available, they could perform RNAscope. The data presented in the manuscript shows that they are very capable of doing this.

Alternatively, or complementarily, they could do transcriptomics with the RNA they obtained from tissues. This might not be as straightforward as it would imply doing experiments (RNAseq) and analyses (bioinformatics) that had not been included in the original manuscript.

**Part III – Minor Issues: Editorial and Data Presentation Modifications**

Reviewer #1: Minor concerns.

1. Fig S2C – please include legends to indicate that they are co-inoculated animals.

2. LN 178 - it is not clear what chicken C15 refers to in Fig 3A and C.

3. In Fig 4 – the quantification of FLAGtag and NP would be a better representation of LP strain than NP+ and HAtag-, as some of the viruses may be defective in expression of viral hemagglutinin protein with HAtag.

4. In LN 383-385 - it is not clear what the authors refer to by “co-sampled airsac”. Does this mean that there was tissue contamination during harvest?

5. In LN394 - HPAIV is naturally selected in 4/6 chickens. The term “naturally” can be misconstrued and should be replaced with HPAIV had a selection advantage over LPAIV or something similar.

Reviewer #2: 6) Were the immunostainings interpreted by a pathologist blind to the treatment group? If so, please specify.

7) For clarity, I would suggest the authors add “H7” in the title.

Reviewer #3: See attached comments.

PLOS authors have the option to publish the peer review history of their article (what does this mean?). If published, this will include your full peer review and any attached files.

Reviewer #1: No

Reviewer #2: No

Reviewer #3: No
---

## [Decision Letter · Decision Letter 1]

11 Dec 2023

Dear Dr. Richard,

Thank you very much for submitting your manuscript "Species-specific emergence of H7 highly pathogenic avian influenza virus is driven by intrahost selection differences between chickens and ducks" for consideration at PLOS Pathogens. As with all papers reviewed by the journal, your manuscript was reviewed by members of the editorial board and by several independent reviewers. The reviewers appreciated the attention to an important topic. Based on the reviews, we are likely to accept this manuscript for publication, providing that you modify the manuscript according to the review recommendations.

Please note minor revisions that include no additional work, just a small description of some of the limitations of the study.

Sincerely,

Daniel R. Perez, PhD

Academic Editor

PLOS Pathogens

Benhur Lee

Section Editor

PLOS Pathogens

Kasturi Haldar

Editor-in-Chief

PLOS Pathogens

orcid.org/0000-0001-5065-158X

Michael Malim

Editor-in-Chief

PLOS Pathogens

orcid.org/0000-0002-7699-2064

Please note minor revisions that include no additional work, just a small description of some of the limitations of the study.

Reviewer Comments (if any, and for reference):

Reviewer's Responses to Questions

**Part I - Summary**

Reviewer #1: I am fine with the authors responses. In lieu of performing additional animal experiments, the authors should consider including a section about the limitations of the study (animal numbers, differences in HPAIV replication in duck vs chickens etc) and indicating that these experiments are designed to be qualitative.

Reviewer #3: This is a submission of a revised manuscript. The authors provided a point by point response to the reviewer's comments. My review of the revised version is described below (see major and minor comments).

**Part II – Major Issues: Key Experiments Required for Acceptance**

Reviewer #1: none

Reviewer #3: The authors have justified why additional *optional* experiments suggested by me are not needed. As they were -in my view- optional I am OK with their decision. Besides, the lack of such additional data does not detract from their findings.

I am in two minds about the request by another reviewer about including more animals: on one hand I see that inter individual variation is expected and for this reason numbers matter (i.e. you need larger numbers to observe clear differences). On the other hand I understand issues about logistics and ethics when using animal species that are not as common as mice.

**Part III – Minor Issues: Editorial and Data Presentation Modifications**

Reviewer #1: none

Reviewer #3: The authors addressed most of the minor issues raised. They also provided proper justification for the minor issues they did not address.

PLOS authors have the option to publish the peer review history of their article (what does this mean?). If published, this will include your full peer review and any attached files.

Reviewer #1: No

Reviewer #3: No

Figure Files:

Data Requirements:

Reproducibility:

References:

---

## [Editor Report · Decision Letter 2]

3 Jan 2024

Dear Dr. Richard,

We are pleased to inform you that your manuscript 'Species-specific emergence of H7 highly pathogenic avian influenza virus is driven by intrahost selection differences between chickens and ducks' has been provisionally accepted for publication in PLOS Pathogens.

Best regards,

Daniel R. Perez, PhD

Academic Editor

PLOS Pathogens

Benhur Lee

Section Editor

PLOS Pathogens

Kasturi Haldar

Editor-in-Chief

PLOS Pathogens

orcid.org/0000-0001-5065-158X

Michael Malim

Editor-in-Chief

PLOS Pathogens

orcid.org/0000-0002-7699-2064
---

## [Editor Report · Acceptance letter]

19 Feb 2024

Dear Dr. Richard,

We are delighted to inform you that your manuscript, "Species-specific emergence of H7 highly pathogenic avian influenza virus is driven by intrahost selection differences between chickens and ducks," has been formally accepted for publication in PLOS Pathogens.

Best regards,

Michael Malim

Editor-in-Chief

PLOS Pathogens

orcid.org/0000-0002-7699-2064